

# Impact of radiation penetration on Antarctic surface melt and subsurface snow temperatures in RACMO2.3p3

Christiaan T. van Dalum[1], Willem Jan van de Berg[1], and Michiel R. van den Broeke[1]

[1]Institute for Marine and Atmospheric Research, Utrecht University, Utrecht, The Netherlands

**Correspondence:** Christiaan van Dalum (c.t.vandalum@uu.nl)

**Abstract.** This study investigates the sensitivity of modeled surface melt and subsurface heating on the Antarctic ice sheet to a new spectral snow albedo and radiative transfer scheme in the Regional Atmospheric Climate Model (RACMO2), version 2.3p3 (Rp3). We tune Rp3 to observations by performing several sensitivity experiments and assess the impact on temperature and melt by changing one parameter at a time. When fully tuned, Rp3 compares well with in situ and remote sensing observations of

surface mass and energy balance, melt, temperature, albedo and snow grain specific surface area. Furthermore, the introduction of subsurface heating in Rp3 significantly improves the snow temperature profile. Near surface snow temperature is especially sensitive to the prescribed fresh snow specific surface area and fresh dry snow metamorphism. These processes, together with the refreezing grain size and subsurface heating, are important for melt around the margins of the Antarctic ice sheet. Moreover, small changes in the albedo and the aforementioned processes can lead to an order of magnitude overestimation of melt, locally

leading to runoff and a reduced surface mass balance.

## 1  Introduction

The contemporary climate of the Antarctic ice sheet (AIS) has been relatively stable, but recently the ice sheet has started losing mass at an accelerated pace (Shepherd et al., 2018). As the AIS contains enough water to raise global mean sea level by 58 m (Fretwell et al., 2013), it is imperative to understand the driving mechanisms behind recent mass loss. Present-day AIS mass

loss has been ascribed to the thinning and breakup of ice shelves, the floating extensions of the ice sheet, due to warming of both ocean and atmosphere (Etourneau et al., 2019). Several Antarctic heat records have been broken in the past decade (Bozkurt et al., 2018), with an all-time record for continental Antarctica of 18.4 °C observed at the tip of the Antarctic Peninsula (AP) in February 2020 (WMO, 2020). These higher temperatures have led to increased surface melt and the formation of melt ponds on the flat ice shelves, enabling the collapse of the Larsen A and Larsen B ice shelves in 1995 and 2002. More ice shelves are

susceptible to collapse if warming continues (Cook and Vaughan, 2010), leading to further AIS mass loss, emphasizing the necessity to fully understand the sensitivity of Antarctic ice shelves to surface melt.

The specific surface mass balance (SMB) of a glacier surface, which is the difference between local accumulation, i.e., mass gain by snowfall and riming, and ablation, i.e., mass loss by runoff, sublimation and drifting snow erosion, is positive for virtually the entire AIS (Agosta et al., 2019; Rignot et al., 2019; Mottram et al., 2020) and only becomes negative in blue ice

areas, where sublimation and erosion exceed snow accumulation (Ligtenberg et al., 2014). The accumulation rate is, however,





also spatially variable, and can be as high as 3 m water equivalent (w.e.) yr$^{-1}$ in the western AP (Van Wessem et al., 2016) and as low as 10 mm w.e. yr$^{-1}$ in the interior of the East Antarctic ice sheet (EAIS) (Van Wessem et al., 2014). For most regions, precipitation dominates the temporal and spatial variability in the SMB signal. Despite low average temperatures (Meyer et al., 2016), significant melt occurs on ice shelves in East Antarctica and the AP (Kuipers Munneke et al., 2012; Lenaerts et al., 2017; Kuipers Munneke et al., 2018). This melt is one to several magnitudes smaller than observed in the western ablation zone of the Greenland ice sheet (Van den Broeke et al., 2016) and almost all meltwater refreezes in the snowpack, or, rarely, is stored englacially (Lenaerts et al., 2017). Consequently, almost no runoff occurs.

Refreezing of meltwater changes the snow structure, as it increases snow grain size. Through large grains, light has to travel a greater distance before it can scatter off a surface, increasing the chance of absorption, thus reducing surface albedo (shortwave reflectivity) (Warren, 2019). This explains the strong snowmelt-albedo feedback, as a lower albedo induces more snowmelt. Jakobs et al. (2019) shows that melt would be three times smaller on an ice shelf in Dronning Maud Land (DML) in East Antarctica without the snowmelt-albedo feedback. Snow grains also increase in size by dry snow metamorphism (Sommerfeld and LaChapelle, 1970), the rate of which increases with temperature. Increasing snow temperature thus means that fresh snow with small grains changes more rapidly into snow with coarser grains, lowering the albedo. With a lower albedo, more energy is absorbed leading to higher temperatures, therefore representing a positive feedback: the dry snow metamorphism-albedo feedback (Picard et al., 2012). Radiation penetration leading to subsurface heating accelerates this process, as subsurface snow is heated more efficiently. The temperature of and melt in the (sub)surface snow of the AIS is thus sensitive to snow grain conditions and subsurface heating. This sensitivity can be investigated locally by using in situ observations, but a polar regional climate model is required to study it for the entire ice sheet.

In this study, we use the polar (p) version of the Regional Atmospheric Climate Model (RACMO2) to analyse the impact of a spectral snow albedo scheme on the (sub)surface temperature and melt of the AIS. RACMO2 has been especially adapted to model glaciated areas (Noël et al., 2018; Van Wessem et al., 2018; Van de Berg et al., 2020; Van Dalum et al., 2021b), and has previously been used to investigate the snowmelt-albedo feedback (Jakobs et al., 2019). Here, we use the latest version, RACMO2.3p3, henceforth Rp3, which has a spectral snow and ice albedo scheme (Van Dalum et al., 2019, 2020) that includes radiation penetration, allowing for subsurface heating and subsurface melt. We evaluate Rp3 with in situ and remote sensing observations, as well as with the previous version, RACMO2.3p2, henceforth Rp2. To investigate the sensitivity of the AIS to (sub)surface heating and snow conditions, we conduct several sensitivity experiments with Rp3, changing one parameter at a time to assess the impact on melt and temperature.

In this manuscript, we first discuss RACMO2 and the sensitivity experiments in more detail in Sect. 2. We also expand upon the concept of SMB, introduce the surface energy balance (SEB) and the observational data sets. Next, results are presented, starting with near-surface and subsurface temperature in Sect. 3, followed by the evaluation of the specific surface area (SSA) in Sect. 4, SEB and albedo in Sect. 5 and SMB in Sect. 6, with a detailed discussion about melt. The results are summarized and conclusions are drawn in Sect. 7.



## 2 Methods and data

### 2.1 Regional climate model

In this study, we use the Regional Atmospheric Climate Model (RACMO2), version 2.3. The model couples the atmospheric dynamics of the High Resolution Limited Area Model, version 5.0.3 (HIRLAM, Undén et al. (2002)) with the atmospheric and surface physics of the European Center for Medium-Range Weather Forecasts (ECMWF) Integrated Forecast System (IFS), cycle 33r1 (ECMWF, 2009) assuming hydrostatic balance. The polar (p) version of RACMO2, developed at the Institute for Marine and Atmospheric Research Utrecht (IMAU), is especially developed for glaciated regions by explicitly modeling snow and ice processes in a dedicated glaciated tile (Noël et al., 2018; Van Dalum et al., 2020).

Dry snow metamorphism in RACMO2 is calculated using the parameterization of the SNICAR snow model (Gelman Constantin et al., 2020), based on the scheme of Flanner and Zender (2006), which considers the impact of temperature, temperature gradient with depth, layer density and initial grain size distribution on grain growth. Based on Eq. (16) of Flanner and Zender (2006), RACMO2 uses the following expression for dry snow metamorphism in meters per time step:

$$\frac{\mathrm{d}r}{\mathrm{d}t} = \frac{\mathrm{d}r}{\mathrm{d}t}\bigg|_0 \left(\frac{\tau}{\tau + 10^6 \cdot (r - \alpha r_0)}\right)^{1/\kappa} \cdot \frac{\Delta t \cdot 10^{-6}}{3600}. \tag{1}$$

With $r$ the grain radius, $r_0$ the initial grain radius, $\frac{\mathrm{d}r}{\mathrm{d}t}\big|_0$ the initial grain growth rate, $\Delta t$ the time step, $\tau$ and $\kappa$ empirical parameters and $\alpha$ a newly introduced tuning parameter that will be changed as an experiment. The grain radius is then converted to SSA using $\mathrm{SSA} = \frac{3}{r \rho_{\mathrm{ice}}}$ (Grenfell and Warren, 1999), with $\rho_{\mathrm{ice}}$ the density of ice, which is set to 917 kg m$^{-3}$ (Bader, 1964). This parameterization uses three regimes based on the initial SSA following observations of Legagneux et al. (2004): 1) for an SSA of 60 m$^2$ kg$^{-1}$ or lower, 2) 60-80 m$^2$ kg$^{-1}$ and 3) 80-100 m$^2$ kg$^{-1}$. Snow metamorphism is fastest for the first regime and slowest for the last. In RACMO2, we assume by default a fresh snow SSA of 60 m$^2$ kg$^{-1}$, hence using the first regime, but this will be changed as a sensitivity experiment.

The latest model version, RACMO2.3p3 (Rp3), includes several updates. The spectrally-integrated (broadband) snow albedo scheme of Gardner and Sharp (2010) is replaced by the Two-streAm Radiative TransfEr in Snow model (TARTES, Libois et al. (2013)). TARTES solves the radiative transfer equation (Jiménez-Aquino and Varela, 2005) by using the Delta-Eddington approximation and geometric-optics Approximate Asymptotic Radiative Transfer (AART) theory (Kokhanovsky, 2004) and provides absorption for each snow layer and spectral albedo for any wavelength between 199 and 3003 nm for both direct and diffuse radiation. It has been coupled to RACMO2 with the Spectral-to-NarrOWBand ALbedo (SNOWBAL) module version 1.2 (Van Dalum et al., 2019). SNOWBAL has been developed to couple the spectral albedos and absorption profiles of TARTES to the 14 narrowbands of the IFS physics scheme in RACMO2 by including albedo and irradiance sub-band variations. The albedo of bands 13 and 14 is almost zero (Gardner and Sharp, 2010), and all radiation in these bands is assumed to be absorbed at the surface. The absorption profiles of TARTES coupled with SNOWBAL now also allows subsurface heating and subsurface melting. Furthermore, a new bare ice albedo scheme has been developed using TARTES and SNOWBAL, but this is of lesser importance for the AIS and is discussed in more detail by Van Dalum et al. (2020).



To properly account for subsurface heating, it has to be considered that heat can still reach the surface within a model time step up to the maximum skin layer equilibration depth (SLED). Between the surface and the SLED, the fraction of shortwave radiation absorbed that attributes to the surface energy balance (SEB) decreases linearly from 1 to 0. The remaining energy contributes to subsurface heating. In other words, the SLED is the maximum depth at which some energy can still equilibrate with the surface within a model time step. Beyond the SLED, all absorbed energy leads to subsurface heating (Van Dalum et al., 2021b).

The multilayer firn module of RACMO2 has also been updated. Numerical diffusion is reduced by a new merging routine that limits the mixing of layers with distinct characteristics. Furthermore, the vertical resolution in snow is increased. Rp3 now typically has 50 to 60 layers, with a maximum of 100. Model output, however, is limited to the upper 20 layers. The impact of the aforementioned model updates for the Greenland ice sheet has been investigated extensively by comparing with in situ and remote sensing measurements (Van Dalum et al., 2020, 2021b), which shows improvements compared to the previous RACMO2 version, Rp2.

## 2.2 Surface mass balance and energy budget

The specific surface mass balance (SMB) represents the net mass gain or loss over a glaciated surface. Some surface processes contribute to mass gain, i.e., snowfall (SN) or rain (RA), and others contribute to mass loss, i.e., sublimation (SU), drifting snow erosion (ER) and runoff (RU). RU includes all liquid water not retained or refrozen in the snowpack. In RACMO2, we adopt the following definition, in kg m$^{-2}$ or mm w.e. yr$^{-1}$:

$$\mathrm{SMB} = \mathrm{SN} + \mathrm{RA} - \mathrm{SU} - \mathrm{ER} - \mathrm{RU}. \tag{2}$$

Formally this definition of the SMB represents the climatic mass balance (Cogley et al., 2011), as internal accumulation, or refreezing, is included.

Melt energy ($M$) is modeled in RACMO2 as the residual energy flux of the SEB of a melting snow or ice surface, with all fluxes in W m$^{-2}$ and defined positive when directed to the surface:

$$M = \mathrm{LW_d} + \mathrm{LW_u} + \mathrm{SW_d} + \mathrm{SW_u} + \mathrm{SHF} + \mathrm{LHF} + G_\mathrm{s}, \tag{3}$$

with $\mathrm{SW_d}$, $\mathrm{SW_u}$, $\mathrm{LW_d}$ and $\mathrm{LW_u}$ the downward and upward shortwave and longwave radiative fluxes, LHF and SHF the turbulent latent and sensible heat fluxes and $G_\mathrm{s}$ the subsurface conductive heat flux. Net shortwave and longwave radiative fluxes ($\mathrm{SW_n}$ and $\mathrm{LW_n}$) are defined as $\mathrm{SW_d} + \mathrm{SW_u}$ and $\mathrm{LW_d} + \mathrm{LW_u}$, respectively. In Rp3, some shortwave radiation is allowed to penetrate through the surface, heating layers below. When snow layer temperature is at melting point, the excess energy is modeled as melt. Percolation of meltwater is modeled using the tipping-bucket method (Coléou and Lesaffre, 1998), where layers are filled with water until the irreducible water content is reached. Any excessive water then percolates to the next unsaturated layer where it can refreeze, run off or be retained by capillary forces, all in a single time step.



**Table 1.** Summary of the RACMO2 sensitivity experiments. No skin layer equilibration depth (SLED) is defined in Rp2.

| Experiment | Fresh snow SSA ($m^2$ $kg^{-1}$) | Snow metam. factor | RF grain size (mm) | SLED (mm) |
|---|---|---|---|---|
| Rp2 | 60 | 1 | 1 | - |
| GRL | 60 | 1 | 1 | 5 |
| FSG | 100 | 1 | 1 | 5 |
| FSM | 100 | 0.25 | 1 | 5 |
| RFG | 100 | 0.25 | 0.25 | 5 |
| CON | 100 | 0.25 | 0.25 | 10 |

## 2.3 RACMO2 experiments

In this study, we perform five sensitivity experiments with Rp3 and compare them to Rp2. All runs are performed on a 27 km grid covering the full AIS with a six minute time step. Radiation and albedo, however, are only calculated on a full-radiation time step, which is every hour. At the boundaries, Rp2 and all Rp3 experiments are forced with 3-hourly ERA5 data (Hersbach et al., 2020). The boundary files include humidity, pressure, temperature and wind speed and direction for each of the 40 atmospheric model layers. The snowpack is initialized by the output of a firn-densification model (IMAU-FDM Ligtenberg et al. (2018)). IMAU-FDM provides the snow grain size, water concentration, temperature, layer thickness and snow and ice density for all initial active layers. No impurities are prescribed in the snowpack, as the impurity concentration of the AIS is typically very low (Warren and Clarke, 1990; Doherty et al., 2010; Dang et al., 2015).

Table 1 summarizes the sensitivity experiments. Rp2 uses the same settings as the Greenland settings in Rp3 (GRL): a fresh snow SSA of 60 $m^2$ $kg^{-1}$, no snow metamorphism tuning, i.e., $\alpha$ in Eq. (1) set to 1, and a refreezing grain size of 1 mm. In GRL, we set the SLED to 5 mm after scale analysis (Van Dalum et al., 2021b). In Rp2, the SLED is not defined, as no radiation penetration occurs and all absorbed shortwave radiation contributes to the SEB.

Four more experiments are performed using Rp3, changing one parameter at a time. In the fresh snow grain size (FSG) experiment, the fresh snow SSA is increased from 60 to 100 $m^2$ $kg^{-1}$, reducing $r$ from 55 µm to 37 µm. An SSA of 100 $m^2$ $kg^{-1}$ better matches observations of fresh snow at Dome C (Libois et al., 2015). Furthermore, this changes the dry snow metamorphism rate from the fastest to the slowest regime, reducing snow growth by an order of magnitude (Fig. 1). This current parameterization, however, is not optimized for Antarctic conditions. Therefore, in the next experiment we reduce fresh dry snow metamorphism (FSM) even more by setting the tuning parameter $\alpha$ in Eq. (1) to 0.25. This reduces fresh snow metamorphism considerably, but its impact diminishes with increasing SSA (Fig. 1). As grain size significantly impacts the albedo (Gardner and Sharp, 2010; He and Flanner, 2020), slower snow metamorphism reduces shortwave radiation absorption in the snowpack, hence snow temperatures are expected to decrease. We also reduce the grain size of refrozen snow from 1 to 0.25 mm (RFG), fitting better with Antarctic observations (Domine et al., 2007), which is expected to further reduce melt. The

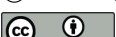

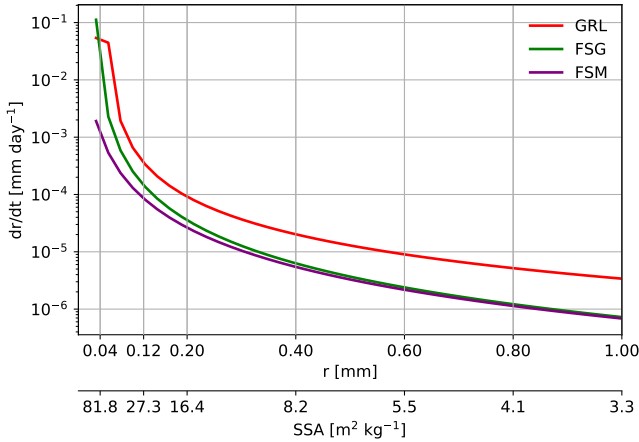

**Figure 1.** Dry snow grain growth as a function of grain radius ($r$) and specific surface area (SSA) for the Rp3 experiments GRL, FSG and FSM.

final experiment is the control run (CON), where the SLED is increased to 10 mm following the scale analysis of Van Dalum
145   et al. (2021b) to better conform to a model time step of 6 minutes. A larger SLED reduces energy available for subsurface heating, further lowering the snowpack temperature.

Running these experiments is computationally demanding, hence only Rp2, GRL and CON are run for the full time period: 1979-2018. FSG, FSM and RFG are run for 1979-1990. For all experiments, 1979-1984 is considered as spin-up. Statistical significance between model versions or observations is determined by using statistical bootstrapping with 2 standard deviation
150   significance.

## 2.4 Observational data

In this study, we use several observational data sets to evaluate the SMB and SEB components, snow and 2-m air temperature, 10-m wind speed and SSA. Here, we provide a brief overview of the observational data sets.

### 2.4.1 SMB

155   Modeled SMB is compared with 1870 SMB measurements including isolated observations and traverses on the EAIS (Fig. 2b). Favier et al. (2013) describe this data set in more detail. In addition, melt fluxes are compared with the output of the surface energy balance model (EBM) of Jakobs et al. (2019). This model is forced with high-quality meteorological and radiation observations to specifically produce a melt rate estimate for Neumayer station (Fig. 2b).





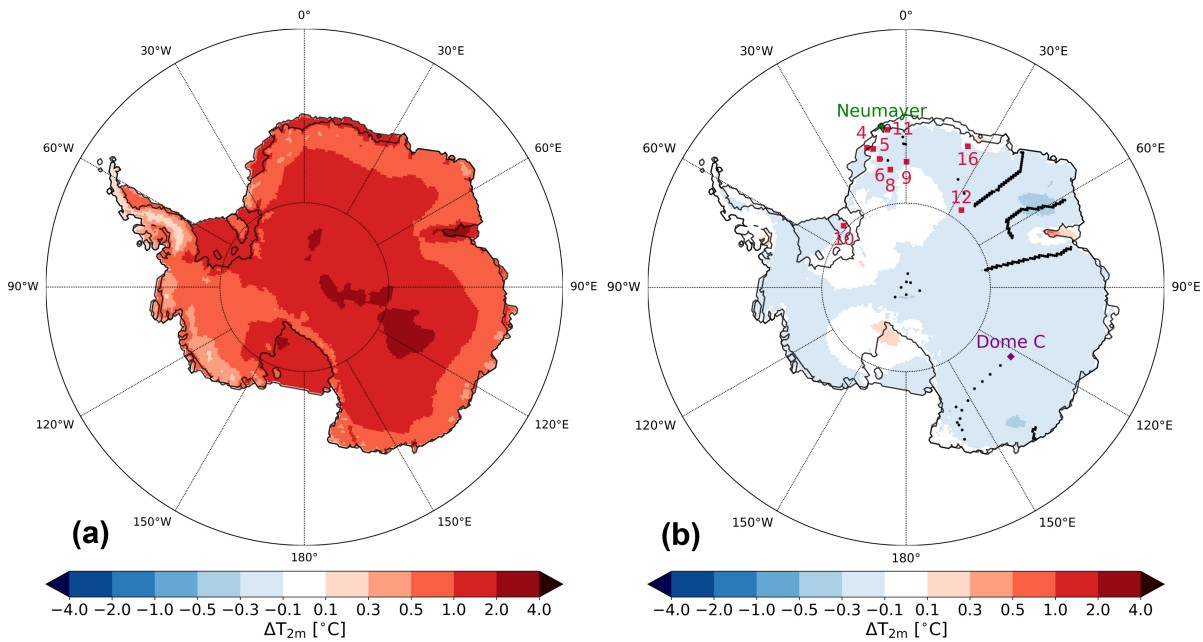

**Figure 2.** Mean yearly-averaged 2-m temperature ($T_{2m}$) difference with Rp2 for **(a)** GRL and **(b)** CON for 1985-2018, with positive values indicating a temperature increase with respect to Rp2. SMB measurement locations are shown in black, AWS in red, Neumayer in green and Dome C in purple.

### 2.4.2 Automatic weather stations

160 The SEB components, 10-m wind speed and 2-m temperature are evaluated using automatic weather station (AWS) data of nine stations, most of them located in DML (Fig. 2b). Some are located on an ice shelf (4, 11) or close to the ice-sheet margin (5, 16) and others are more in-land, hence covering several climatic regimes. All data are monthly averaged. Van Wessem et al. (2018) provide a more detailed overview of the AWS specifications.

### 2.4.3 QuikSCAT melt fluxes

165 The time series of the satellite radar backscatter from the SeaWinds scatterometer aboard QuikSCAT (QSCAT) is used to produce a seasonal meltwater product covering the entire AIS (Trusel et al., 2013). This melt product uses an empirical relation between the satellite product and in situ observations. The QSCAT melt product is provided on a 4.45 km resolution grid, but is resampled to the RACMO2 grid with the nearest neighbor method. Here, we use QSCAT to evaluate the modeled ice sheet wide surface meltwater fluxes between 2000-2009.





**Table 2.** Statistics of the monthly-averaged downward, upward and net longwave and shortwave fluxes during summer ($LW_d$, $LW_u$, $LW_n$, $SW_d$, $SW_u$, $SW_n$, respectively), albedo, sensible heat flux (SHF), latent heat flux (LHF), 2-m temperature ($T_{2m}$), skin temperature ($T_{skin}$) and 10-m wind speed ($V_{10m}$) using AWS data of DML between 1997 and 2012 (locations shown in Fig. 2b). We use the ratio of the monthly sum of $SW_u$ and $SW_d$ to determine the albedo. For all variables, 202 observations are available. The correlation coefficient ($R^2$), bias and root-mean-square error (RMSE) are shown for Rp2, GRL and CON. In all following figures, Rp2 is in black, GRL in red and CON in blue.

| Variable | Unit | Rp2 | | | GRL | | | CON | | |
|---|---|---|---|---|---|---|---|---|---|---|
| | | $R^2$ | bias | RMSE | $R^2$ | bias | RMSE | $R^2$ | bias | RMSE |
| $LW_d$ | W m$^{-2}$ | 0.94 | -10.8 | 14.8 | 0.93 | -3.4 | 12.3 | 0.94 | -11.4 | 15.3 |
| $LW_u$ | W m$^{-2}$ | 0.96 | 7.5 | 10.6 | 0.97 | -4.8 | 7.9 | 0.97 | 7.9 | 9.7 |
| $LW_n$ | W m$^{-2}$ | 0.63 | -3.3 | 9.6 | 0.56 | -8.3 | 12.8 | 0.66 | -3.5 | 9.3 |
| $SW_d$ | W m$^{-2}$ | 0.93 | 9.4 | 25.0 | 0.92 | 9.0 | 25.7 | 0.93 | 15.5 | 26.7 |
| $SW_u$ | W m$^{-2}$ | 0.94 | -14.0 | 21.8 | 0.93 | -1.2 | 17.4 | 0.95 | -20.6 | 25.6 |
| $SW_n$ | W m$^{-2}$ | 0.69 | -4.6 | 12.5 | 0.61 | 7.8 | 15.5 | 0.73 | -5.1 | 11.9 |
| Albedo | - | 0.26 | 0.018 | 0.03 | 0.21 | -0.020 | 0.04 | 0.39 | 0.022 | 0.03 |
| SHF | W m$^{-2}$ | 0.58 | 5.9 | 8.1 | 0.60 | 2.4 | 5.9 | 0.62 | 6.5 | 8.4 |
| LHF | W m$^{-2}$ | 0.73 | 2.7 | 3.5 | 0.66 | 0.3 | 2.6 | 0.72 | 2.9 | 3.7 |
| $T_{2m}$ | °C | 0.98 | -0.3 | 1.4 | 0.97 | 2.0 | 2.7 | 0.98 | -0.8 | 1.6 |
| $T_{skin}$ | °C | 0.98 | -1.4 | 2.0 | 0.97 | 1.6 | 2.3 | 0.98 | -1.9 | 2.4 |
| $V_{10m}$ | m s$^{-1}$ | 0.16 | -1.9 | 2.4 | 0.19 | -2.2 | 2.7 | 0.20 | -1.8 | 2.3 |

### 2.4.4 Subsurface snow temperature

Snow temperatures of Rp3 are compared to temperature probe measurements that provide hourly snow temperatures at various depths at Dome C during December 2006 (Fig. 2b) (Brucker et al., 2011). Probes are positioned down to 21 m depth, but we limit the evaluation to the upper 2 m. Temperatures are measured every 10 cm starting between 10 and 60 cm depth, and every 20 cm between 80 and 200 cm.

### 2.4.5 SSA

The SSA of the upper snow layers at Dome C are retrieved by Picard et al. (2016) between 2013 and 2015 by using an algorithm applied to observed spectral albedos. This SSA product is representative for the upper two centimeters, as the albedo for such a vertically homogeneous snow layer, with an SSA of 50 m$^2$ kg$^{-1}$ or larger, is representative for more than 95% of the observed surface albedo (Fig 1. of Picard et al., 2016). Measurements are available between September and March.



**Figure 3.** Mean yearly-averaged 2-m temperature ($T_{2m}$) difference with Rp2 for **(a)** GRL, **(b)** FSG, **(c)** FSM, **(d)** RFG and **(e)** CON for 1985-1990. The dots represent statistical significance.

## 3 Results: Temperature

Figure 2 shows the yearly-averaged $T_{2m}$ difference for GRL and CON with Rp2. Considerably higher temperatures are simulated in GRL, with some areas more than 2.0°C warmer with respect to Rp2. The temperature in CON (Fig. 2b) is on average only 0.1 to 0.3 °C lower than Rp2. In summer (not shown), the signal of Fig. 2 is amplified. A comparison with observations in DML during summer (Table 2), which is the season where any changes in the albedo have the strongest impact on the SEB, shows that the temperature of Rp2 is modeled well, with a small bias of -0.3°C and a root-mean-square error (RMSE) of 1.4°C. The bias of GRL and CON are larger: 2.0°C and -0.8°C respectively. This illustrate the high sensitivity to the implemented changes on the $T_{2m}$ for the AIS in RACMO2. The new radiative transfer scheme results in a lower albedo, which is especially important during summer and will be discussed in more detail in Sect. 5. Including radiation penetration leads to higher subsur-

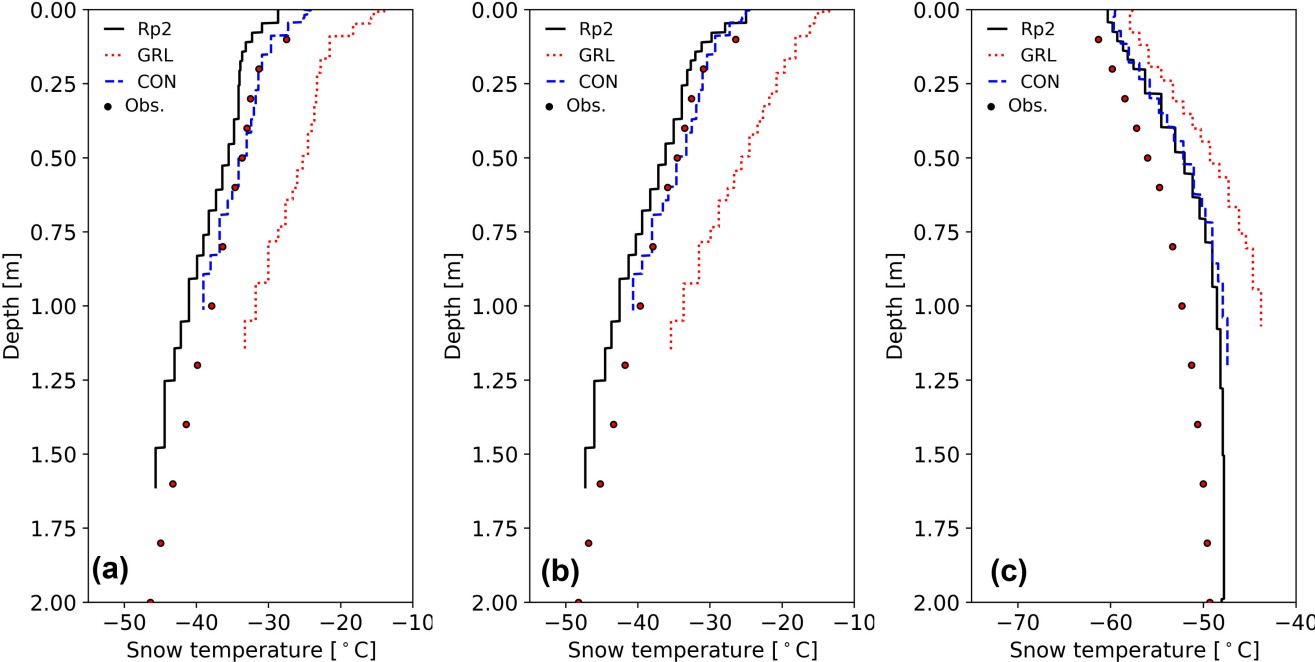

**Figure 4.** Subsurface snow temperature profile for Dome C for the 20 upper snow layers of Rp2, GRL and CON and observations (Obs.) for **(a)** 5 January 2007, **(b)** 17 January 2007 and **(c)** 5 April 2007, all measured at 06:00 UTC (14:00 LT).

face snow temperatures, enhancing snow metamorphism and subsequently enhancing radiation absorption. Due to this positive

feedback, inaccuracies in the modeled (sub)surface snow metamorphism (Flanner and Zender, 2006) are amplified in Rp3.

To investigate the exact cause of deviating temperatures, we show the yearly-averaged $T_{2m}$ difference with Rp2 for all sensitivity experiments for 1985-1990 (Fig. 3). As Rp2 models $T_{2m}$ fairly well, differences with this model version cannot be too large and it is used as a benchmark. All implemented changes lower the temperature, although some changes impact it more than others. A significant lowering of the temperature is induced by the increase of the fresh snow SSA to 100 m$^2$ kg$^{-1}$

in the FSG experiment (Fig. 3b). FSG also uses a different fresh snow regime in the grain growth parameterization (Sect. 2.1, Fig. 1) and grains with a high SSA consequently remain at the surface for longer.

The strongest temperature lowering occurs when we further reduce the fresh dry snow metamorphism (Fig. 3c) by implementing a tuning parameter (Eq. (1)). As Fig. 1 illustrates, this tuning reduces in particular the snow metamorphism for small grains, i.e., up to 100 times slower metamorphism in FSM than FSG. This tuning makes that surface layers with a high

SSA (>50 m$^2$ kg$^{-1}$) are more persistent between snow deposition events, consequently lowering the surface temperature and hence, through turbulent and longwave exchange between the surface and near-surface atmosphere, reducing $T_{2m}$. The significant temperature differences between Fig. 3a and Fig. 3c shows how sensitive RACMO2 is to grain size and underlines the importance of an accurate snow metamorphism scheme.



Higher temperatures are relatively persistent on some of the ice shelves (Fig. 3c), especially in DML. These regions are
characterized by melt in summer that refreezes in the snowpack. As meltwater refreezes, it increases snow grain size, resulting
in more solar radiation absorption and therefore higher temperatures. Reducing the refreezing snow grain size consequently
reduces the temperature difference on relatively dry locations with melt (Fig. 3d). Increasing the SLED further lowers the
temperature as subsurface heating is reduced (Fig. 3e). The temperature in CON is now somewhat too low during summer (Tab
2). This bias can be further reduced by slightly changing $\alpha$ in Eq. (1).

## 3.1 Snow temperature

An important addition in Rp3 is subsurface penetration of shortwave radiation, which allows subsurface absorption and local
heating of the snowpack. For Greenland, Van Dalum et al. (2021b) show that Rp3 models higher subsurface snow temperatures,
as a result of internal heating, that match well with observations at Summit. In the ablation zone, the melting point is reached
to a greater depth than in Rp2, enabling subsurface melt. Here, we show that the snow temperatures at Dome C (Fig. 4) match
better with observations (Brucker et al., 2011) in CON than in Rp2 and GRL. During summer (Fig. 4a and b), we observe
that Rp2 is somewhat too cold compared to measurements. Results improve for CON, showing the significance of subsurface
heating, although the skin temperature in DML is somewhat underestimated (Table 2). Figure 4a and b also show that in
GRL, i.e., without tuning, the snow temperatures are significantly overestimated by up to 10 °C. During autumn (Fig. 4c),
temperature profiles of Rp2 and CON, and to a lesser extent GRL, are more similar, as the impact of radiation penetration
diminishes towards winter. Compared to observations, however, temperatures in autumn are too high.

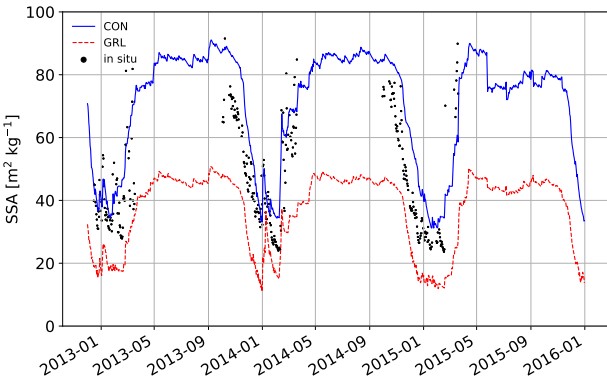

**Figure 5.** Time series of average SSA for Dome C of the upper two centimeters of the snowpack in CON, GRL and as observed by Picard
et al. (2016).



## 4 Specific surface area comparison

In the previous section, we illustrated the importance of grain size on the temperature of the AIS. Compared to in situ observations at Dome C (Picard et al., 2016), the SSA of the upper two centimeters in the CON simulation follows the yearly cycle well (Fig. 5). The SSA drops gradually over time during spring and summer to values around 40 m$^2$ kg$^{-1}$, which is
somewhat higher than observed. In GRL, the SSA is too low as it drops below 20 m$^2$ kg$^{-1}$. The SSA decline during spring is delayed by a few weeks, but the rate of change is similar to observations. After summer, the SSA gradually increases with deposition of fresh snow, but only reaches 40 to 50 m$^2$ kg$^{-1}$ for GRL, significantly below observations. For CON, the SSA gradually increases to 80 to 90 m$^2$ kg$^{-1}$, which is in agreement with observations. Note that the average SSA of the upper two centimeters never reaches the prescribed fresh snow SSA of 100 m$^2$ kg$^{-1}$, as large snowfall events at this polar desert site
are rare (Picard et al., 2019). To summarize, the GRL settings lead to unrealistically low SSAs. The CON settings somewhat underestimate snow metamorphism, leading to higher SSA during summer, but this can be fine tuned using $\alpha$ in Eq. (1).

## 5 Surface energy balance and albedo analysis

Table 2 shows the statistics of SEB components compared to AWS observations in summer from DML in Rp2, GRL and CON. All fluxes toward the surface are defined positive.
The longwave radiation of Rp2 and CON correlate well with observations, but some biases are observed. The underestimation of LW$_d$ illustrates that the atmosphere in RACMO2 is too cold. This could be due to too few clouds, too low atmospheric humidity or biases in the radiation scheme for these cold conditions. This is partly compensated by underestimated LW$_u$, resulting in a relatively small LW$_n$ bias. In GRL, the bias of LW$_n$ is larger, as higher surface temperatures lead to an overestimation of LW$_u$, while only partly compensated by increased LW$_d$.
Table 2 shows that SW$_d$ is overestimated for all model experiments. As no parameters that directly impact SW$_d$ have been changed, it illustrates that the atmosphere is too transparent, likely due to similar reasons as causing the LW$_d$ differences. For Rp2 and CON, this bias is compensated by SW$_u$, as the albedo is somewhat too high during summer. Table 2 also shows that the albedo of GRL is on average too low, which is discussed in more detail in Sect. 5.1, resulting a lower compensating SW$_u$ and consequently a larger SW$_n$ bias.
On average, the SHF is overestimated in RACMO2 during summer, despite an underestimation of the wind speed ($V_{10m}$, Table 2). This can be either due to an incorrect representation of the roughness length or an incorrect temperature gradient between surface and atmosphere. In GRL, turbulent heat exchange is smaller while $T_{2m}$ is overestimated. For a stable surface layer, this therefore suggests that the temperature of lower atmospheric layers is too high in RACMO2. Similarly, GRL also shows a better LHF representation than Rp2 and CON. Hence, turbulent fluxes can still be further improved.





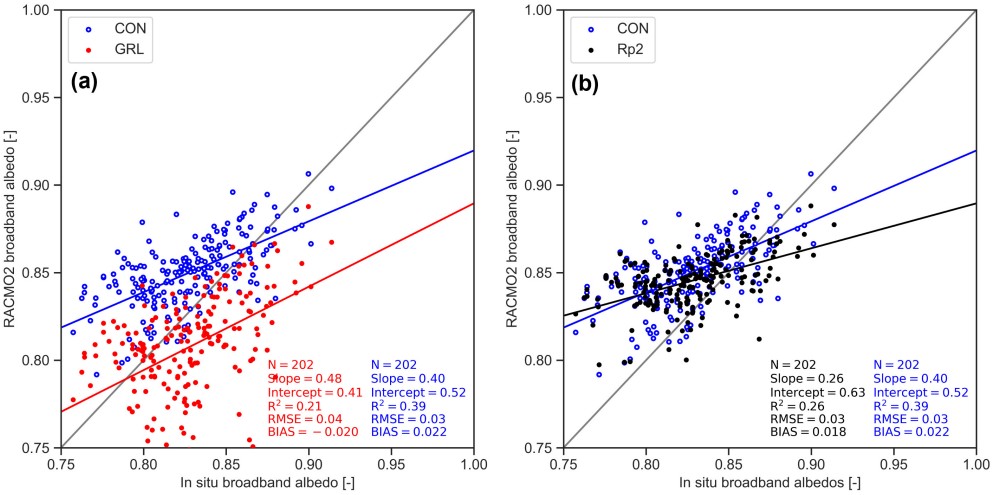

**Figure 6.** Monthly-averaged albedo in DML in **(a)** CON and GRL and **(b)** CON and Rp2 compared to AWS measurements between 1997 and 2012. The gray lines are the 1-on-1 lines and the red and blue lines are linear regression of the data, with N the number of observations, the slope, the intercept, $R^2$ the correlation coefficient, the bias and root-mean-square error (RMSE).

## 5.1 Albedo


Year-round monthly-averaged albedo in DML compared to observations is shown in Fig. 6. Figure 6a illustrates that the spread in data points in GRL is similar to CON but with a lower average. Moreover, an albedo lower than 0.8 is sometimes modeled in GRL and is shown by observations, while absent in Rp2 and CON (Fig. 6b).

Yearly averaged, the albedo of CON is relatively homogeneous over the AIS (Fig. 7a) with a high albedo (> 0.8) almost
everywhere due to the abundance of fine-grained snow. Compared to Rp2, the differences are generally small, with somewhat higher albedos in West Antarctica (Fig. 7c). The albedo of GRL is significantly smaller than Rp2 (Fig. 7b), showing the impact of snow properties on the radiative transfer scheme in Rp3. The largest differences in both GRL and CON are observed for the Amery ice shelf, where bare ice can be found at the surface during summer. The transition from snow to bare ice is faster due to higher snow temperatures, leading to more snow-free days and consequently a lower mean albedo. Note that the albedo
in Rp2 is fixed for bare ice, while TARTES and SNOWBAL are called in Rp3, allowing a variable ice albedo depending on atmospheric conditions (Van Dalum et al., 2020).

## 5.2 Neumayer case study

Figure 8 shows a case study at Neumayer for the one-year period July 2012 to July 2013 at local noon, illustrating the various processes that impact the albedo on seasonal and sub-seasonal time scales. In general the albedo is high (close to 0.9, Fig.
8b) but fluctuating, mostly depending on cloud cover (Fig. 8f). The albedo is on average lower than the broadband albedo parameterization of Gardner and Sharp (2010) (G&S, Fig. 8c) employed in Rp2. Simulating radiation penetration by applying



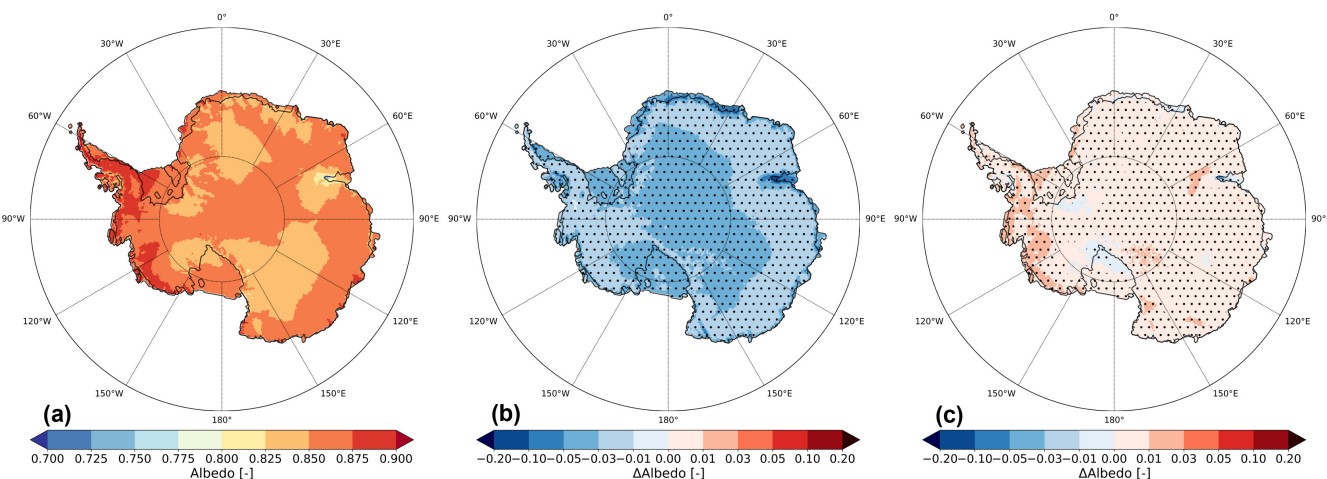

**Figure 7. (a)** Mean yearly-averaged albedo in CON and albedo difference with Rp2 in **(b)** GRL and **(c)** CON for 1985-2018. The dots represent statistical significance.

a simple exponential decay function with depth for radiation to G&S, as Kuipers Munneke et al. (2011) (PKM) did, lowers the albedo, reducing the difference with CON. This illustrates the importance of radiation penetration even with the presence of fresh snow during most months (Fig. 8g). The removal of fresh snow by sublimation during summer does not lead to

considerable differences with G&S and PKM. The addition of a thin snow layer (only millimeters thick) on top of firn in February, on the other hand, induces a strong albedo increase, resulting in a large albedo difference of more than 0.1 with PKM (Fig. 8d). Such differences reduce over time when snow metamorphism occurs or if more fresh snow is deposited. This illustrates that a simple exponential decay function is not enough to properly capture radiation penetration.

The impact of cloud cover on irradiance is shown in Fig. 8a. It shows that infrared (IR) radiation is filtered out by clouds,

but that cloud content (Fig. 8f) is too small to considerably impact UV and visible irradiance. As the spectral albedo of IR radiation is low (Dang et al., 2015; Warren, 2019), the broadband albedo in Rp3 consequently increases with increasing cloud content. Compared to G&S and PKM, cloud cover induces stronger albedo variations in CON, as this effect is now explicitly taken into account.

Solar zenith angle (SZA) also impacts the albedo. The albedo increases with SZA, as it increases the angle of incidence of

radiation, leading to a higher likelihood for light to scatter out of the snowpack (Solomon et al., 1987; Gardner and Sharp, 2010). The spectral distribution of light also changes with increasing SZA. For high SZA ($>80°$), a relatively larger part of the irradiance is IR (Fig. 8a), for which the spectral albedo is low, partly compensating the albedo increase. This effect, however, is not captured in G&S and PKM, but is included in Rp3. Consequently, the albedo is lower for CON for high SZA, as can be seen at the beginning of May during clear-sky conditions (Fig. 8c, d).

Compared to observations, the daily mean albedo product of CON is often too high (Fig. 8h), especially during spring and summer, while the albedo of GRL is often too low during summer and too high during spring. To summarize, tuning the snow

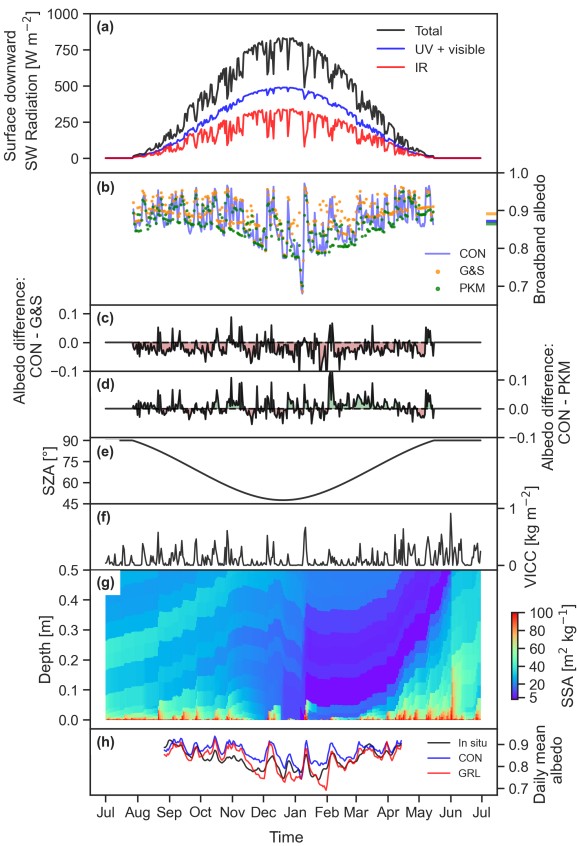

**Figure 8.** Time series at Neumayer for 2012-2013, 12:00 UTC (12:00 LT). **(a)** Instantaneous surface downward shortwave radiation, split into infrared (IR), and ultraviolet (UV) and visible radiation; **(b)** instantaneous broadband albedo for CON, the parameterization of Gardner and Sharp (2010) (G&S) and Kuipers Munneke et al. (2011) (PKM). The horizontal lines on the right indicate the mean. **(c)** Albedo difference CON - G&S and **(d)** CON - PKM; **(e)** solar zenith angle (SZA); **(f)** vertically integrated cloud cover (VICC), which is the summation of the liquid and ice water path; **(g)** SSA as a function of depth and **(h)** daily mean albedo of CON, GRL and in situ observations.

layers to better fit with SSA observations (Fig. 5) and temperature (Fig. 3) does not necessarily lead to a smaller bias in the SEB components or albedo. The analysis of the SEB shows that RACMO2 has some compensating biases, i.e., clouds and turbulence. Nonetheless, despite a slightly overestimated albedo, CON provides a better representation of the near-surface
climate, albedo and near-surface snow state than the GRL experiment.

## 6 Surface mass balance

Figure 9 shows the mean yearly-accumulated SMB, melt, precipitation and sublimation difference with Rp2 for GRL (upper row) and CON (lower row). In CON, the SMB differences are generally small (lower than 10 mm w.e. yr$^{-1}$), with somewhat

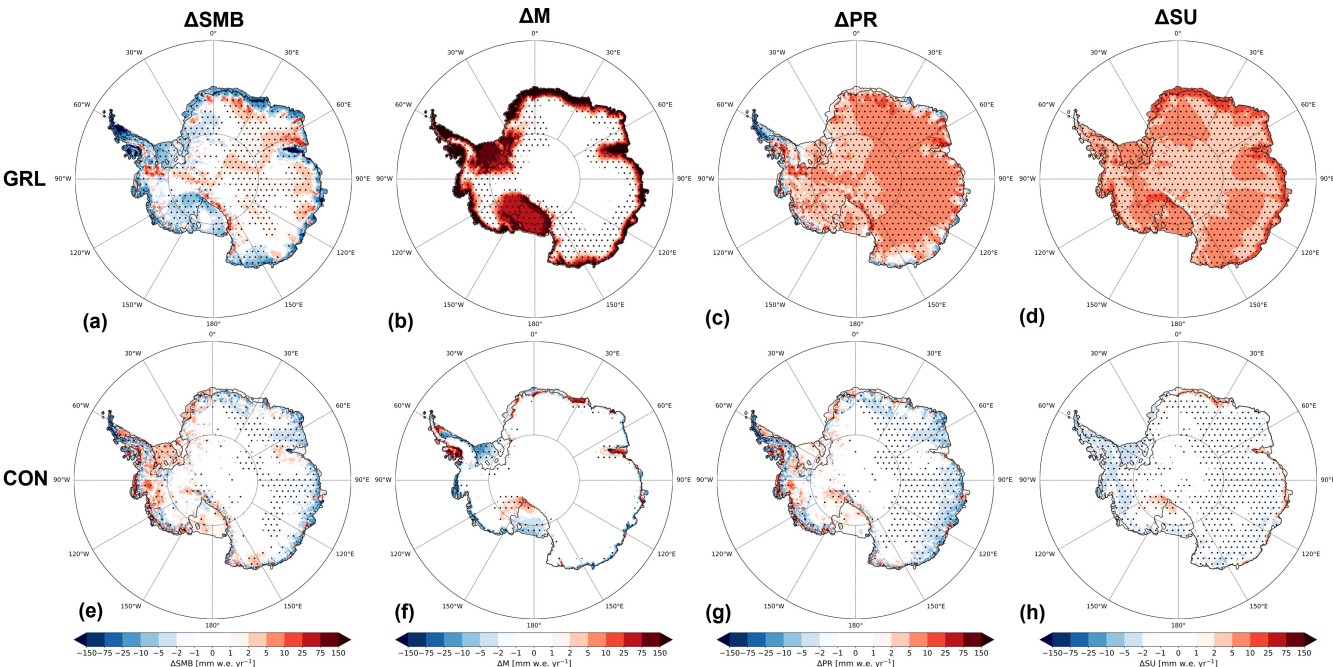

**Figure 9.** Yearly accumulated SMB, melt (M), precipitation (PR) and sublimation (SU) difference with Rp2 for GRL (**(a)**-**(d)**, respectively) and CON (**(e)**-**(h)**, respectively) for 1985-2018, with positive values showing an increase with respect to Rp2. Runoff and drifting snow erosion are not shown. The dots represent statistical significance.

larger differences for the West Antarctic ice sheet (WAIS) and the AP that are driven by precipitation changes. The precip-
itation changes are minor, however, as total precipitation for the WAIS and AP are more than an order of magnitude larger
(Van Wessem et al., 2016). Melt has increased on the Wilkins, George VI and northern part of the Larsen C ice shelf in the AP
and the Amery ice shelf in East Antarctica. The changes of Rp3 are therefore largest for warm regions where melt is already
significant, in agreement with Van Dalum et al. (2020, 2021b). Runoff, however, remains limited (not shown) and almost all
meltwater is buffered in the snowpack where it refreezes. Only at the southern part of the Amery ice shelf is the retention
capacity now exceeded and runoff modeled, hence lowering the SMB. The margins of DML show considerable year-to-year
and spatial melt variability. This demonstrates the high sensitivity of the implemented changes for this region, as the snowpack
is close to the melting point in summer and additional energy absorption therefore leads to a stronger meltwater flux.

In GRL (upper row of Fig. 9), a strong SMB decrease is modeled for ice shelves in the AP, DML and Amery ice shelf.
More inland, the SMB increases somewhat, which is mainly caused by an ice sheet-wide precipitation increase. It is, however,
partially compensated by more sublimation. As the precipitation parameterization has not been changed, the moisture of this
excess precipitation has been taken up locally. Further analysis showed that it relates to unrealistic features during summer in
GRL. Due to the higher surface temperature, sublimation increases and a cloud-topped shallow convective layer is modeled for
the interior of the ice sheet. These clouds subsequently provide the additional precipitation. This synoptic weather pattern is,





**Figure 10.** Mean yearly-accumulated melt difference with Rp2 in mm w.e. $yr^{-1}$ for **(a)** GRL, **(b)** FSG, **(c)** FSM, **(d)** RFG and **(e)** CON for 1985-1990. Positive values show a melt increase with respect to to Rp2. The dots represent statistical significance.

however, not backed by observations. Furthermore, melt has increased strongly around the margins of the entire AIS and all ice
shelves. This melt changes the snow structure and leads to extensive runoff on several smaller ice shelves in DML, where the
snowpack is close to saturation in summer, and on the Amery, Larsen C, Wilkins and George VI ice shelves. Finally, compared
to 1870 SMB observations in the EAIS (locations shown in Fig. 2b), the difference between CON and GRL is small and both
agree well with measurements (Fig. A1), with a bias of 15.0 and 15.6 mm w.e. $yr^{-1}$, RMSE of 80.64 and 82.27 mm w.e. $yr^{-1}$
and correlation coefficient of 0.45 and 0.44, respectively.

**6.1   Melt**

To investigate what causes the strongly overestimated melt in GRL, Fig. 10 shows the melt difference with Rp2 for all sensitivity
experiments. By increasing the fresh snow SSA (Fig 10b) and reducing fresh dry snow metamorphism (Fig 10c), less radiation



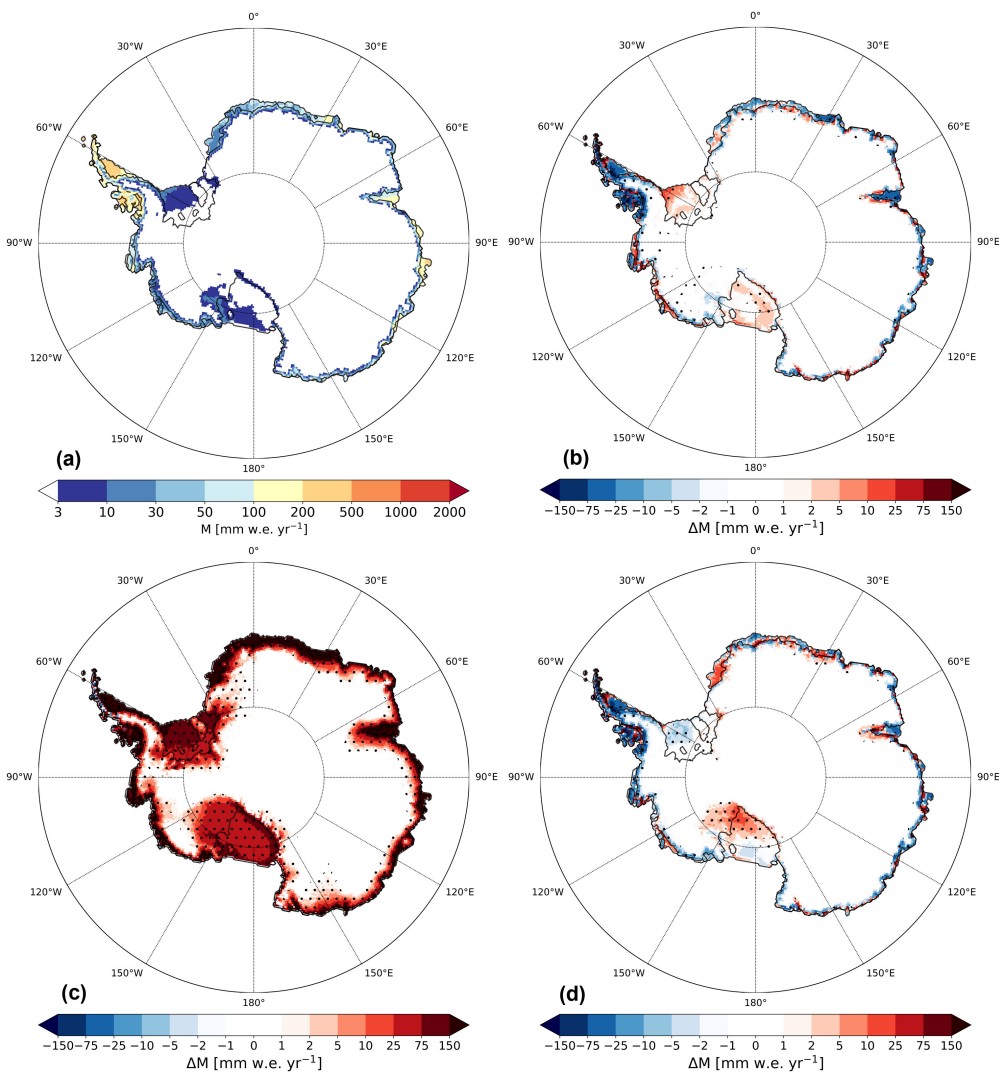

**Figure 11. (a)** Mean yearly-averaged melt in mm w.e. yr$^{-1}$ estimated by QSCAT, melt difference with QSCAT for **(b)** Rp2, **(c)** GRL and **(d)** CON for 2000-2009. For **(b)**-**(d)**, positive values indicate a melt increase with respect to QSCAT. The dots represent statistical significance.

is absorbed, lowering melt for most regions, in particular for the Ross and Filchner-Ronne ice shelves. These ice shelves are covered by fine-grained snow for most of the year and are therefore especially sensitive to changes in the fresh snow parameterization. The change to the fresh snow SSA and metamorphism delays the onset of the melt season, but its impact diminishes as the melt season progresses. Unsurprisingly, a strong melt reduction occurs by lowering the refreezing grain size (Fig 10d), which leads to less energy absorption in areas with refreezing. For the time step currently employed in Antarctic simulations, a SLED of 5 mm leads to slightly overestimated heat buffering in the uppermost part of the snow layer, leading to more internal melt. Hence, melt is further reduced by increasing the SLED. Integrated over the AIS, yearly-averaged melt has





increased by as much as 490% in GRL with respect to Rp2. Each sensitivity experiment lowers the melt flux, resulting in only a 7.0% increase in CON (Table A1). As a result, the domain-integrated yearly-averaged SMB modeled in GRL is lower (2370 Gt yr$^{-1}$) than CON (2407 Gt yr$^{-1}$).

### 6.1.1 Comparison with QuikSCAT

In this section we compare modeled melt with QuikSCAT (QSCAT) data (Sect. 2.4.3). QSCAT shows that virtually no melt
occurs on the majority of the AIS (Fig. 11) and that there are only small melt fluxes (< 100 mm w.e. yr$^{-1}$) around most of the margins of East and West Antarctica. More melt is observed in the AP, especially on the ice shelves, but it is still one order of magnitude smaller than observed in the ablation zone of west Greenland (Van den Broeke et al., 2016).

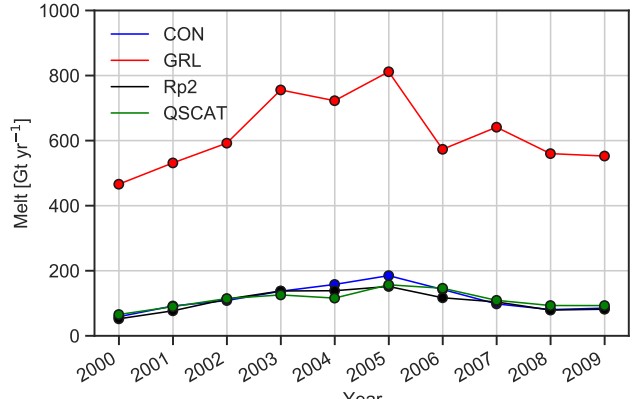

**Figure 12.** Domain-integrated yearly-averaged melt for the AIS in Gt yr$^{-1}$ measured by QSCAT and modeled in Rp2, GRL and CON.

Compared to QSCAT, Rp2 (Fig. 11b) and CON (Fig. 11d) perform generally well, with small differences around the margins of the WAIS and EAIS. In the AP, Rp2 and CON predict more melt in the northern part of Larsen C, while melt is underes-
timated in the southern part. Furthermore, melt in the western AP appears underestimated. For the Wilkins and George VI ice shelves, however, CON models higher melt fluxes compared to QSCAT, similar to Fig.9f. The melt in GRL (Fig. 11c) is overestimated by more than an order of magnitude for almost all regions close to the ice-sheet margin. Furthermore, GRL models a relatively large melt flux for the Ross and Filchner-Ronne ice shelves, where QSCAT observes virtually no melt.

Integrating melt over the AIS shows a similar pattern (Fig. 12), with melt in GRL almost an order of magnitude larger than
QSCAT in every year. This shows the impact of internal heating and melt-albedo feedback, as these processes significantly enhance melt, similar to findings of Jakobs et al. (2019). CON and Rp2 correlate better with QSCAT. The interannual variability, however, correlates well for all experiments.



### 6.1.2 Comparison with an EBM

Figure 13 shows the cumulative melt at Neumayer station as calculated by the EBM of Jakobs et al. (2019), which is forced
by meteorological data, and compares it with Rp2, GRL and CON. Also for this location, GRL predicts excessively high melt.
This figure confirms that GRL significantly overestimates melt and that tuning is necessary. CON initially underestimates melt,
which is compensated by increased meltwater production in the warm years of 2004, 2010 and 2014, ending closer to the
cumulative melt of the EBM than Rp2.

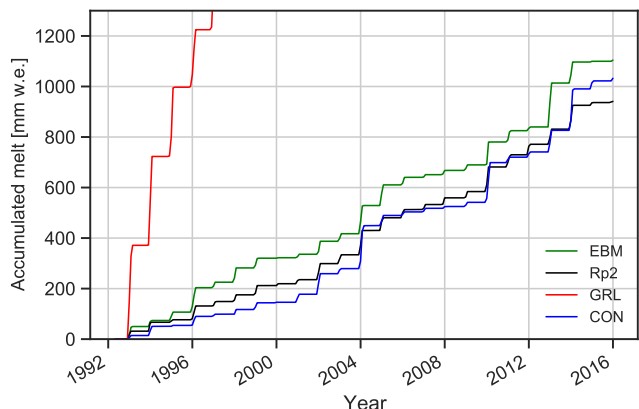

**Figure 13.** Cumulative melt in mm w.e. at Neumayer calculated by the energy balance model (EBM) of Jakobs et al. (2019) and Rp2, GRL
and CON.

In conclusion, melt of the AIS is somewhat sensitive to fresh snow SSA and fresh dry snow metamorphism and is highly
sensitive to the refreezing grain size and SLED. Hence, subsurface heating can warm the snowpack considerably, enhancing
melt. Despite the low average melt flux in Antarctica, the impact of subsurface heating should not be neglected for a physical
description of (sub)surface melt.

## 7 Summary and conclusions

In this study, we investigated the impact of a new snow albedo and radiative transfer scheme in the latest version of RACMO2
on the near-surface temperature, subsurface snow temperature, SMB, SEB, albedo and melt of the AIS. We tuned Rp3 by
changing one parameter at a time, allowing us to investigate the sensitivity of the AIS to each change.
We have run Rp3 for the entire AIS on a 27 km grid forced at the boundaries by 3-hourly ERA5 data. Three experiments are
run for the full period (1979-2018): Rp2, Rp3 with Greenland (GRL) settings and the Rp3 control run (CON) that includes all
tuning steps. The results are compared to in situ and remote sensing observations and to the previous model version Rp2. The
other sensitivity experiments are done for 1979-1990 and include increasing the fresh snow SSA (FSG), reducing the fresh dry
snow metamorphism (FSM) and lowering the refreezing grain size (RFG).



Compared to observations and Rp2, the 2-m temperature in the GRL experiment is considerably higher. The sensitivity experiments show improvements, resulting in a lower bias with observations for CON. The reduction of the fresh dry snow metamorphism rate in the FSM experiment results in a lowering of the temperature. For some areas, however, the 2-m temperature is now somewhat too low. This indicates, together with SSA observations, that the fresh dry snow metamorphism might have been reduced too strongly and that further improved results would likely be reached with a larger value for $\alpha$ of Eq. (1). More importantly, the results presented here highlight the necessity to correctly model snow conditions, and that the current snow metamorphism scheme has to be improved or replaced. Nonetheless, subsurface temperatures in CON have improved significantly and match very well with measurements, showing the added value of subsurface heating to model performance.

Analysis of the SEB shows that RACMO2 exhibit some small (max. 10 W m$^{-2}$) persistent biases in the net radiative fluxes, which is caused by too transparent clouds and overestimated turbulent surface fluxes. These biases demonstrate that Rp2 and CON provide a better representation of the surface climate than GRL, and, moreover, that there is room for model development of RACMO2.

The higher (subsurface) temperatures in GRL lead to excessive melt around the margins and on the ice shelves, locally leading to runoff and a reduced SMB. Integrated over the AIS, melt in GRL is one order of magnitude larger than observed by QSCAT and also considerably larger than measured at Neumayer station. In contrast, CON and Rp2 compare well with these observations. Melt is progressively reduced by all sensitivity experiments, especially in RFG and CON, showing the sensitivity of the AIS to the refreezing grain size and SLED. It is clear that GRL does not produce realistic meltwater fluxes and that the standard Greenland settings of Rp3 should not be used for the AIS. This is undesirable, as model settings should preferably not depend on location and/or tuning to local conditions, and shows that more research into this problem is needed.

In conclusion, introducing a more physically based albedo scheme in RACMO2 that allows for radiation penetration and subsurface heating improves, after tuning, subsurface snow temperatures in Antarctica. Incorrectly modeling snow conditions can lead to an order of magnitude melt overestimation and can significantly impact the climate and lower the SMB of the AIS. Furthermore, as is shown in the GRL experiment, only a small lowering of summer albedo by, for example, global warming induced melting would lead to a very different near-surface summer climate in Antarctica.

*Author contributions.* CTvD, WJvdB and MRvdB initiated this study and analyzed the results. CTvD lead the writing of the manuscript, performed the simulations and implemented model changes. All authors contributed to the discussion on the manuscript.

*Data availability.* Data are available at 27 km resolution for Antarctica for CON and GRL (1979-2018) and FSG, FSM and RFG (1979-1990). Monthly-accumulated and monthly-averaged data for $T_{2m}$ and SMB and its components are available for all Rp3 experiments. SEB components are available for GRL and CON. The data can be found here: https://doi.org/10.5281/zenodo.5512077 (Van Dalum et al., 2021a). Rp2 data are available form the authors.



*Competing interests.* The authors declare that they have no conflict of interest.

*Acknowledgements.* We acknowledge financial support from the European Union's Horizon 2020 research and innovation programme (PRO-TECT) under agreement number 869304, PROTECT contribution number TBD. We also acknowledge the Netherlands Earth System Science
Centre and the ECMWF for computational time on their supercomputers.



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



**Appendix A**

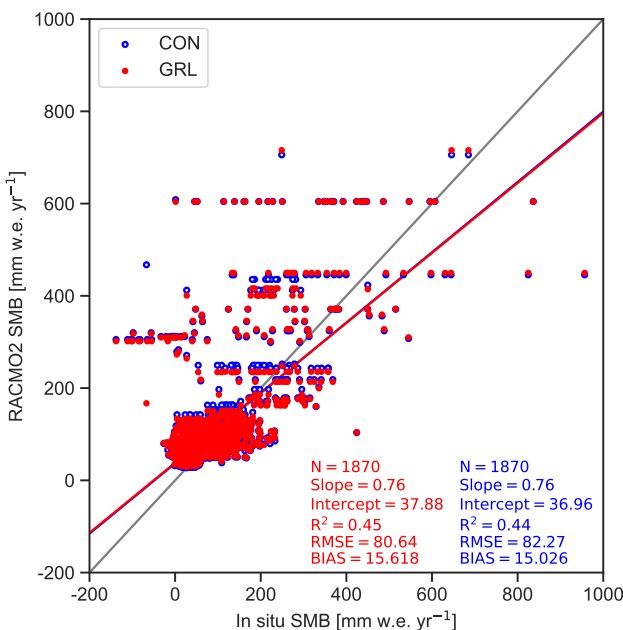

**Figure A1.** Yearly accumulated SMB in the EAIS in CON and GRL compared to observations. The gray line is the 1-on-1 line and the red and blue lines are linear regression of the data, with N the number of observations, the slope, the intercept, $R^2$ the correlation coefficient, the bias and root-mean-square error (RMSE). The intercept, bias and RMSE are in mm w.e. $yr^{-1}$.

**Table A1.** Domain-integrated yearly-averaged SMB and melt for the AIS in Gt $yr^{-1}$ for Rp2 and the Rp3 sensitivity experiments for 1985-1990. For both variables, the difference with Rp2 in percentage is also shown.

| Experiment | SMB (Gt $yr^{-1}$) | $\Delta$SMB (in %) | Melt (Gt $yr^{-1}$) | $\Delta$Melt (in %) |
|---|---|---|---|---|
| Rp2 | 2422 | | 115 | |
| GRL | 2370 | -2.1 | 679 | +490 |
| FSG | 2381 | -1.7 | 469 | +307 |
| FSM | 2390 | -1.3 | 351 | +205 |
| RFG | 2402 | -0.8 | 183 | +59 |
| CON | 2407 | -0.6 | 123 | +7.0 |