# Peer review of "Sensitivity of Antarctic surface climate to a new spectral snow albedo and radiative transfer scheme in RACMO2.3p3"

_The Cryosphere, 2021_

## Referee Comment (RC1)

**Impact of radiation penetration on Antarctic surface melt and subsurface snow temperatures in RACMO2.3p3**
By Van Dalum et al 2021

Van Dalum et al. updates in the subsurface scheme in RACMO2.3p3, by setting up different experiments where they change one parameter at the time. They go into depth on how the changes in the different parameters impact the albedo, melt and temperature of the snow. They compare the different experiments with an older version of RACMO and in situ data. They find that the albedo and melt are very sensitive to the parameterization of the subsurface, especially over the ice shelves.

Overall, I find this to be a very nicely written paper, with interesting and important results. However, some comments should be addressed before publication.

**Major comment:**

It is sometimes hard to figure out what version of RACMO you refer to, e.g. when you write RACMO2 is it then always the polar version or always the non-polar version, and in P3L67, you mention RACMO2 but which version. This is mainly a problem in sections 1 and 2, it decreases the readability of these sections. Please be more clear on this!

**Minor comments:**

**General comment:**

In some places such as P11L216-218 and P13L256 you use the wording significance/significantly, whereas, in other places, such as caption in Fig 3 and 7, you use the wording statistical significance. Are there differences between these two wordings? Have you performed a significance test for both kinds of wordings? Generally, the use of the word significant implies that the significance has been tested

In some of the section titles you abbreviate and in others there are no abbreviations, like section 2.2 Surface mass balance and energy budget and section 2.4.1 SMB, or in section 2.4.5 SSA and 4 Specific surface area comparison, please be consistent.

I find the use of the experiment name "Greenland settings/GRL" a bit misleading. I get what you mean, but there were several times when I read the manuscript where I thought of the Greenlandic ice sheet when I read GRL. I suggest that you consider changing this experiment name.

**Specific comments:**

P1L2: Is it not the same "2" in RACMO2 and in version 2?, so the second "2" is redundant? "Regional Atmospheric Climate Model (RACMO2), version 2.3p3"

P2L45-53: Specify which time period you are studying

P2L56: Can you please defined what you mean by specific surface area (SSA)

P3L61: Same it P1:L2 comment, maybe just write RACMO2.3 since the RACMO abbreviation is already introduced in P2L45

P3L61: Maybe just write RACMO2.3 since the RACMO2 abbreviation is already introduced on P2L45

P3L61: Should there not be a "p" for polar in version 2.3?

P3L73: Is the alpha solely introduced to tune for this experiment? If so, maybe change the sentence to "and α is a newly introduced tuning parameter to make changes in this study/experiment"

P3L79-90: You write "The latest model version, RACMO2.3p3 (Rp3), includes several updates." But as far as I understand you are only describing the TARTES scheme and how it works. That is only one update, right, not several? And then in P4L91 you talk about the SLED update, and finally in P4L97 talking about the layer update. Maybe, list all three updates in P3L79 and then go into details afterward, then it is easier for the reader to keep track of the updates.

P4L98-99: You state that the vertical resolution is increased, and Rp3 has 50-60 layers, does that mean the layer thickness decreased, or is it the number of layers that have increased?

P4L98-99: Is the number of layers constant over time and over the domain?

P4L99-101: Sentence is a bit hard to understand, maybe put in some commas like:
"The impact of the aforementioned model updates, has for the Greenland ice sheet, been investigated extensively by comparing with in situ  and remote sensing measurements

P4L106: Is that the polar version of RACMO2?

P4L107: $kg\ m^{-2}$ should have a time unit when mm w.e has

P4L119: Just out of curiosity, (you do not have to state this in your manuscript) is the run-off time slope dependent, i.e is the run-off time the same over steep and flat terrain? and what happens if a layer refreezes and thus cannot be penetrated, is the water then treated as run-off?

P7L167: What timely resolution of the QSAT product do you use?

P8L173: Why do you limit the evaluation to the first 2 meters? Is that equivalent to the upper 20 layers in the model?

P10L193-194: What is meant by the sentence "differences with this model version cannot be too large and it is used as a benchmark"

P10L194: What is meant by the sentence " All implemented changes lower the temperature," is far as I read/understand figure 3, experiment a and b gets warmer and c, d, and e looks to become colder.

P11L212: showed instead of show

P17L315: Consider merging section 6 Surface mass balance and 6.1 melt into one, since melt is discussed in both.

P19L329: Just use QSCAT, since you already introduced this abbreviation in section 2.4.3

P20L349-352: This statement belongs in section 7

P21L364-365: The sentence "For some areas, however, the 2-m temperature is now somewhat too low." does not sound like a conclusion, try to quantify the bias and rephrase

P25L480: The Mottram et al paper, is no longer in The Cryosphere Discussion, but in The Cryosphere

**Figure comments:**

Figure 4: In the legend the Obs dot is shown it full black, whereas on the plots it is shown as a red dot with a black outer edge, please change it to the same color.

Figure 4: In the caption, change the date to  5$^{th}$ of January or January 5$^{th}$ and the same for the other dates

Figure 7a: Please use a sequential color map instead of diverging

Figure 8: Please make it wider, it is hard to see the timely fluctuations

Figure 9: Can you make higher values in the colorbar, it looks like Delta melt for GRL is much higher than 150 mm per year

Figure 10: Same comment as Figure 9

Figure 11: Same comment as Figure 9

Figure 11a: Please use a more sequential color map

---

## Author Comment (AC1)

Referee comment responses on the manuscript:
*Impact of radiation penetration on Antarctic surface melt and subsurface snow temperatures in RACMO2.3p3*
by C.T. van Dalum et al.

We would like to thank the referees for their comments and we address them here. In black the comment, in orange the response, in blue the changes that we would implement in the manuscript. All page numbers refer to the old manuscript version. See also the marked-up version for all other minor changes, like some fixed typos, implemented in the manuscript.

After considering all comments, we decided to change the title to:
**'Sensitivity of Antarctic surface climate to a new spectral snow albedo and radiative transfer scheme in RACMO2.3p3 '**

**Review #1**

Major comment:
It is sometimes hard to figure out what version of RACMO you refer to, e.g. when you write RACMO2 is it then always the polar version or always the non-polar version, and in P3L67, you mention RACMO2 but which version. This is mainly a problem in sections 1 and 2, it decreases the readability of these sections. Please be more clear on this!
We agree that usage of the name RACMO is sometimes confusing. We have changed it throughout the manuscript to Rp3, except for the following lines where we have changed it to:
P2 L46:
The polar version of RACMO2 has been especially...
P2 L54:
...we first discuss Rp2, Rp3 and the sensitivity...
P3 L66 added the following:
Here, we present the latest model version, RACMO2.3p3 (Rp3).
P3 L67:
Dry snow metamorphism in both the previous version, RACMO2.3p2 (Rp2), and Rp3 is calculated using the...
P3 L70
Rp2 and Rp3 use the...
P3 L77
A fresh snow SSA of 60 m2 kg-1 is used in preceding RACMO studies, hence using the first regime, but this will be changed as a sensitivity experiment.
P3 L79
Rp3 includes several updates. The spectrally-integrated...
P4 L101
...which shows improvements compared to Rp2.
P4 L106
In Rp2 and Rp3, we adopt...
P5 L121, the title of the subsection:
RACMO2.3p3 experiments
P12 L236
...that the atmosphere is too cold in the model.
P12 L249
...atmospheric layers is too high in the model.
P15 L288
that there are some compensating biases

P20 L354
…in the latest adaptation of the polar version of RACMO2 on the near-surface temperature…
P21 L381
…based albedo scheme in the polar version of RACMO2 that allows…

Minor comments:
General comment:
In some places such as P11L216-218 and P13L256 you use the wording
significance/significantly, whereas, in other places, such as caption in Fig 3 and 7, you use
the wording statistical significance. Are there differences between these two wordings? Have
you performed a significance test for both kinds of wordings? Generally, the use of the word
significant implies that the significance has been tested
We agree that this is confusing. For all occurrences of the word 'significant', the significance has been
tested. So, using the word 'statistical' is indeed redundant and has been removed in the manuscript.

In some of the section titles you abbreviate and in others there are no abbreviations, like
section 2.2 Surface mass balance and energy budget and section 2.4.1 SMB, or in section
2.4.5 SSA and 4 Specific surface area comparison, please be consistent.
This is indeed inconsistent and we changed the following titles:
2.4.1 Surface mass balance
2.4.5 Specific surface area
6.1.2 Comparison with an energy balance model

I find the use of the experiment name "Greenland settings/GRL" a bit misleading. I get what
you mean, but there were several times when I read the manuscript where I thought of the
Greenlandic ice sheet when I read GRL. I suggest that you consider changing this
experiment name.
As requested, we changed the name GRL to GS throughout the manuscript and the figures.
Additionally, we have changed the following
P5 L130
…the sensitivity experiments. The settings of the first Rp3 experiment, the Greenland settings
experiment (GS), are the same as used for investigating the Greenland ice sheet by Van Dalum et al.
(2021). Rp2 uses the…

Specific comments:
P1L2: Is it not the same "2" in RACMO2 and in version 2?, so the second "2" is redundant?
"Regional Atmospheric Climate Model (RACMO2), version 2.3p3"
Yes, it is redundant and we have changed it to:
Furthermore, we changed this throughout the manuscript
P1L2:
Regional Atmospheric Climate Model (RACMO), version 2.3p3

P2L45-53: Specify which time period you are studying
Adjusted as requested
P2L50
We evaluate Rp3 with in situ and remote sensing observations, as well as with the previous version,
RACMO2.3p2, henceforth Rp2, between 1979 and 2018.

P2L56: Can you please define what you mean by specific surface area (SSA)
We changed the following to address this:

P2L56
… followed by the evaluation of the specific surface area of snow, defined as the total surface area per kilogram, in Sect. 4…
P3L73
The grain radius is then converted to specific surface area (SSA)…

P3L61: Same it P1:L2 comment, maybe just write RACMO2.3 since the RACMO abbreviation is already introduced in P2L45
P3L61
In this study, we use the regional climate model RACMO2.3.

P3L61: Should there not be a "p" for polar in version 2.3?
The 'p' is introduced shortly after.

P3L73: Is the alpha solely introduced to tune for this experiment? If so, maybe change the sentence to "and α is a newly introduced tuning parameter to make changes in this study/experiment"
Yes this is introduced only for this experiment.
P3L73
The tuning parameter α is added in Rp3.

P3L79-90: You write "The latest model version, RACMO2.3p3 (Rp3), includes several updates." But as far as I understand you are only describing the TARTES scheme and how it works. That is only one update, right, not several? And then in P4L91 you talk about the SLED update, and finally in P4L97 talking about the layer update. Maybe, list all three updates in P3L79 and then go into details afterward, then it is easier for the reader to keep track of the updates.
You are right that it is confusing. We have updated the first line of the paragraph to clarify this.
P3L79
Rp3 includes several updates. A new snow and ice albedo scheme has been introduced, subsurface heating is now accounted for and improvements have been made to the multilayer firn module, including changes to the merging and splitting routine of snow layers.

P4L98-99: You state that the vertical resolution is increased, and Rp3 has 50-60 layers, does that mean the layer thickness decreased, or is it the number of layers that have increased?
In general, the mean layer thickness has decreased as there are now more layers in the upper snow layers. This also results in a higher number of layers.
P4L99
…with a maximum of 100, resulting in more layers in the upper snowpack.

P4L98-99: Is the number of layers constant over time and over the domain?
No it is dynamic, both in space and time.
P4L98
Furthermore, the vertical resolution in snow is increased, resulting in more layers near the surface. The number of layers is dynamic, Rp3 now typically has 50 to 60 layers, with a…

P4L99-101: Sentence is a bit hard to understand, maybe put in some commas like:
"The impact of the aforementioned model updates, has for the Greenland ice sheet, been investigated extensively by comparing with in situ and remote sensing measurements
We have changed it to the following:
P4L99

The impact of the aforementioned model updates has been investigated extensively for the Greenland ice sheet, by comparing with in situ and remote sensing measurements…

P4L106: Is that the polar version of RACMO2?
See previous comments

P4L107: kg m⁻² should have a time unit when mm w.e has
You are right:
P4L107
$kg\ m^{-2}\ yr^{-1}$

P4L119: Just out of curiosity, (you do not have to state this in your manuscript) is the run-off time slope dependent, i.e is the run-off time the same over steep and flat terrain? and what happens if a layer refreezes and thus cannot be penetrated, is the water then treated as run-off?
In RACMO, run-off is instantaneous, so there are no differences between a steep or flat slope. For runoff, there are no ice lenses, i.e., even if a layer has a high density and might be interpreted as an ice layer, meltwater is still able to percolate through it.

P7L167: What timely resolution of the QSAT product do you use?
For QSCAT, we use a seasonal, i.e., yearly, meltwater product, as is explained in P7L166

P8L173: Why do you limit the evaluation to the first 2 meters? Is that equivalent to the upper 20 layers in the model?
We limit the evaluation to 2 meters because the upper 20 model layers are always within this depth for both Rp2 and Rp3 (See Fig. 4). We have added the following to clarify this:
P8L172:
Probes are positioned down to 21 m depth, but as the upper 20 model layers are always located within 2 m, we limit the evaluation to this depth.

P10L193-194: What is meant by the sentence "differences with this model version cannot be too large and it is used as a benchmark"
To clarify this, we changed the following:
P10 L193
As Van Wessem et al. (2018) have shown that Rp2 models T2m fairly well, it is therefore used as a benchmark.

P10L194: What is meant by the sentence " All implemented changes lower the temperature,"
is far as I read/understand figure 3, experiment a and b gets warmer and c, d, and e looks to become colder.
You are right that figure 3 a and b show that the temperature is too high. What we meant to say here is that with each subsequent change, the temperature is lowered with respect to the previous experiment.
P10L194
Similar to Fig 2., the temperature of GS is overestimated significantly. All subsequently implemented changes lower the temperature, although some changes impact it more than others.
P10L196
…at the surface for longer. The temperature, however, is still too high.

P11L212: showed instead of show
Adjusted

P17L315: Consider merging section 6 Surface mass balance and 6.1 melt into one, since melt is discussed in both.

This is a good suggestion. We merged 6 and 6.1 and renamed it 'Surface mass balance and melt'. We also renamed 6.1.1 to '6.1 Melt comparison with QuikSCAT' and 6.1.2 to '6.2 Melt comparison with an energy balance model'.

P19L329: Just use QSCAT, since you already introduced this abbreviation in section 2.4.3

Adjusted

P20L349-352: This statement belongs in section 7

We have removed this part.

P21L364-365: The sentence "For some areas, however, the 2-m temperature is now somewhat too low." does not sound like a conclusion, try to quantify the bias and rephrase

We changed the following

P21 L364

For some areas, however, the 2-m temperature is now too low. Yearly-averaged, it is underestimated by up to 0.5 degrees C.

P25L480: The Mottram et al paper, is no longer in The Cryosphere Discussion, but in The Cryosphere

Correct

Figure comments:

Figure 4: In the legend the Obs dot is shown it full black, whereas on the plots it is shown as a red dot with a black outer edge, please change it to the same color.

Adjusted as requested

Figure 4: In the caption, change the date to 5th of January or January 5th and the same for the other dates

Done

Figure 7a: Please use a sequential color map instead of diverging

We tried to apply a continuous sequential map (on the left). Indeed, spatial gradients are easier to see, however we find that it is more difficult to read the actual values. Therefore, we decided to keep the current color scheme. We did change the range from 0.7 to 0.9, to 0.8 to 0.9, which enhances the amount of detail that is displayed:

[Figure]

Figure 8: Please make it wider, it is hard to see the timely fluctuations
Adjusted

Figure 9: Can you make higher values in the colorbar, it looks like Delta melt for GRL is much higher than 150 mm per year
Done

Figure 10: Same comment as Figure 9
Done

Figure 11: Same comment as Figure 9
Done

Figure 11a: Please use a more sequential color map
Similar to Fig 7a: We tried to apply a continuous sequential map (on the left), but we find that it is much more difficult to read the actual melt values, especially in regions with low melt rates, like DML. In the sequential map, it is hard to see a difference between 3, 20 or 50 mm w.e. yr-1 of melt, which are the typical melt rates of most of the Antarctic ice shelves. Therefore, we decided to keep the current color scheme. We did change the color range to better accommodate for the melt that occurs on Antarctica, i.e., we limited the max melt plotted to be 400 mm w.e. yr-1. The new figure is on the right.

[Figure]

**Review #2**

Line 24: Please update the Mottram et al. reference.
Correct

Line 56: An explanation of what the 'specific surface area' is would be useful here, as a non-expert may not be familiar with this term.
We changed the following to address this
P2L56 … followed by the evaluation of the specific surface area of snow, defined as the total surface area per kilogram, in Sect. 4…
P3L74 The grain radius is then converted to specific surface area (SSA) , defined as the total surface area per kilogram, …

Line 67: Please explain what SNICAR stands for.
P3L67
…using the parameterization of the Snow, Ice, and Aerosol Radiative (SNICAR) model…

Line 72-78: I'm not clear what the parameter tau is. What are its dimensions? Based on the definition of SSA then it should be meters? The parameter tau is usually reserved for a timescale, so this is confusing. You also mention snow metamorphism is fastest for the first regime and slowest for the last, so this again suggests a timescale. Please clarify in the manuscript. (After reading further, is this symbol actually the radius r? This is never defined.)
We agree that it is confusing that we chose the 'tau' for this parameter, as this parameter is indeed in meters and it is not a timescale. It is simply an empirical parameter for grain size evolution. We have changed it to beta. For the three regimes, the tuning parameters are chosen in such a way that it results in fast, intermediate and slow metamorphism.
P3L72
With r the grain radius, r0 the initial grain radius, dr/dt_0 the initial grain growth rate, delta_t the time step, and beta (in m) and kappa are empirical parameters for grain size evolution. The tuning parameter alpha is added in Rp3.

Line 91-92: This sentence does not make sense. As written, you are saying that the model time step is set equal to a depth. Also, can you explain what SLED actually means?
This paragraph is indeed somewhat vague and we reformulate it. We meant to say that by introducing radiation penetration, subsurface heating occurs. However, one has to realize that in reality heat conduction to the surface on time scales smaller than a model time step occurs. Energy that is absorbed close to the surface can still equilibrate with the surface within a model time step and hence escape via longwave radiation emission or turbulent fluxes. This near-surface subsurface heating thus contributes to the surface energy balance. The maximum depth that some energy can still equilibrate on the time scale of a model time step is then what we call the maximum skin layer equilibration depth (SLED). Beyond the SLED, we can safely assume that heat cannot equilibrate with the surface within a model time step and therefore entirely contributes to subsurface heating. So, this also means that the SLED depends on the chosen time step, as heat located at a larger depth may still equilibrate with the surface for a larger time step. As is stated in the text, we then assume that the fraction of shortwave radiation absorbed that attributes to the SEB is 1 at the surface and decreases linearly to 0 at the SLED. If this process is neglected, the near surface heat conduction is underestimated, leading to overestimated subsurface temperatures and hence melt.
P4L91
Not all shortwave radiation absorbed in the snowpack leads effectively to subsurface heating. Close to the surface, absorbed heat can diffuse and therefore equilibrate with the surface on time scales

shorter than a model time step. With increasing depth, an increasingly larger part of subsurface shortwave radiation is unable to equilibrate with the surface and is therefore attributed to subsurface heating. The maximum depth that some energy can still equilibrate with the surface within a model time step is what we call the maximum skin layer equilibration depth (SLED). Beyond this depth, all energy contributes to subsurface heating. Between the surface and the SLED, the fraction of shortwave radiation absorbed that attribute to the SEB decreases linearly from 1 to 0 (illustrated in Fig. 1 of Van Dalum et al. (2021)). In other words, a larger SLED means that a larger fraction of shortwave radiation entering the snowpack contributes to the SEB and subsurface heating is therefore reduced. If the SLED is chosen too small, near subsurface heating is overestimated.

Line 105: Can rain contribute to mass gain? Do you mean via refreezing? Rain could melt ice/snow as well as runoff itself.
Yes, rain contributes to mass gain (See Eq. (2)). This water can be retained or refreeze within the snowpack and is treated similarly as meltwater, so it can also percolate. The addition of rainwater will not lead to melting in RACMO. This rain-induced melting is a rather small effect, and cannot be easily implemented in RACMO as the temperature of rain is not modelled in RACMO.
P4L118
Percolation of rain and meltwater is modeled…

Line 126: Need a comma or semi-colon before the reference.
Adjusted as requested

Line 148: What criteria did you use to assess that spin-up had been reached?
In RACMO2, we use the temperature and snowpack conditions for spin-up. To limit the impact of the initial snowpack, we run the model for a few years, which allows now snow layers to form. After about five years, there has been enough snowfall to form a significant snowpack even for the driest locations.
P6L148:
For all experiments, 1979-1984 is considered as spin-up, as this time is required to build up a proper snowpack required for the albedo calculations and to limit the impact of initialization on the temperature.

Line 158: What is the justification for looking at Neumayer? Are these results representative of the wider Antarctic region?
As Neumayer is located on the ice shelf, it is representative for the more vulnerable areas of Antarctica to melt, i.e., the ice shelfs. And as we show in the manuscript, the ice shelves are in particular tricky to model correctly, especially in East Antarctica, of which Neumayer is part of. Furthermore, the observation record at Neumayer is one of the longest and best documented time series of its kind. Therefore, investigating the location of Neumayer station helps in understanding the ice shelves. We added:
P6L158
Neumayer station is representative for ice shelves surrounding the EAIS, as it is located on one of them.

Figure 2: What are the numbers in this figure?
The numbers represent AWS stations.
P7 Fig. 2:
… locations are shown in black, the numbered AWS stations in red, Neumayer…

Abstract and in the text : "We tuned Rp3 by changing one parameter at a time, allowing us to investigate the sensitivity of the AIS to each change." : You did not change one parameter at a time, you changed the parameters incrementally. It would have been good to change one parameter at a time to decipher more clearly the role of each of your tuning parameter. E.g. you advocate along the article that "subsurface heating" is important, but in your experiments, parameters affecting subsurface heating (SLED, and maybe RF grain size?) are used on top of the others, so that at the end it is difficult to compare their relative importance.

You are right that we changed the parameters incrementally. We have changed the text accordingly. We chose to incrementally change the parameters and show the differences that way. By doing it this way, we are able to show what the actual impact is of the implemented changes on the end product, which is what we are most interested in. It is true that by not incrementally changing the parameters that one can investigate the exact impact that one parameter has in theory, but as the combination of all changes are implemented, we do not think that it is a fair assessment. Some changes will have diminishing returns on each other, for example the FSG and FSM changes, as they both impact the metamorphism rate. The impact that changing only one parameter is therefore not the impact that it would have on the end product. Last but not least, running RACMO2 is computationally expensive, and also running the model for an extensive amount of time for each separate change would result in considerably more required computational power.

P1L4:

… and melt by incrementally changing one parameter at a time.

P2L52:

… we conduct several sensitivity experiments with Rp3 by incrementally changing one parameter at a time…

P5L134:

Four more experiments are performed using Rp3, incrementally changing one parameter at a time.

P20L355:

We tuned Rp3 by incrementally changing one parameter at a time

Abstract : "Furthermore, the introduction of subsurface heating in Rp3 significantly improves the snow temperature profile."

Conclusion : "Nonetheless, subsurface temperatures in CON have improved significantly and match very well with measurements, showing the added value of subsurface heating to model performance."

I think there is not enough evidence in the article to support this statement (detailed below).

We agree that this conclusion is not sufficiently justified (see also below) and we have rewritten the following:

P1L5

… and energy balance, melt, near-surface and (sub)surface temperature, albedo and snow grain specific surface area. Near surface snow temperature is…

P21L368

…be improved or replaced. Nonetheless, subsurface temperatures of CON match well with observations at Dome C for the summer of 2007.

Finally, I have an open question: with regard with your evaluation, it seems that at the end, you don't improve the albedo compared to Rp2, and you obtain very similar bias in CON and in Rp2. Is it a compromise to obtain similar results as in Rp2 for other variables?

You are right that despite the introduction of many new physical processes in the new snow albedo scheme, that the broadband albedo for much of the AIS is similar to Rp2. We do show in Fig 8,

however, that with the introduction of new physics the albedo changes quite drastically on a day-to-day basis. Many of these changes, however, average out on a monthly scale. So the albedo of Rp2 is often right for the 'wrong' reason. Furthermore, the parameterization of Rp2 is designed specifically to do well for fresh snow conditions. As fresh snow conditions are the most dominant on Antarctica, it is not surprising that it generally does rather well. Also, the impact of the new albedo scheme of Rp3 is largest on warmer conditions with some melt, as is often the case in Greenland. As it is considerably warmer in the GS setting, the albedo is therefore also lower (see Fig 6a). But these lower values disappear in CON as the temperature is generally much lower. We have added the following:
P14L284:
…clear-sky conditions (Fig. 8c, d). Monthly-averaged, however, the aforementioned processes have a limited effect, as most differences between CON and Rp2 are averaged out (Fig. 6b).
P21L373
With the introduction of a new physically based albedo and radiative transfer scheme, more processes now impact the snow albedo. Radiation penetration and spectral shifts due to cloud cover and high SZA can lead to albedo differences up to -0.1 between CON and Rp2. Monthly-averaged, however, differences between these model versions are small.

Along the text :
p1 L20 Cook and Vaughan, 2010 : there is much recent litterature on this topic
Done, we have added citations to: Trusel et al. (2015) and Martin et al. (2019)

p1 L22-23 "mass gain by snowfall and riming" : and drifting snow transport ?
P1 L23:
…i.e., mass gain by snowfall, riming and drifting snow accumulation, and ablation, i.e., mass loss by runoff, sublimation and drifting snow erosion, …

p2 L26-27 "can be as high as 3 m water equivalent (w.e.) yr−1 in the western AP (Van Wessem et al., 2016) and as low as 10 mm w.e. yr−1 in the interior of the East Antarctic ice sheet (EAIS) (Van Wessem et al., 2014)" : it would be better to cite observational article here instead of again model output paper.
As requested, we cited a paper of Picard et al. 2019 that measured the accumulation rate at DomeC in East Antarctica. In the manuscript of Van Wessem et al. (2016) they not only model the SMB, but they also use a large SMB observations data set. The largest SMB observed is more than 3000 mm w.e. yr-1. In the caption of Fig. 3 of Van Wessem et al. (2016): "Two locations with either observed or modelled SMB values > 3000 mm we yr-1", indicating that there is an observation of more than 3 m w.e. yr-1. We rephrased this sentence to the following:
P2L26
The accumulation rate is, however, also spatially variable, and is measured to be as high as 3 m water equivalent (w.e) yr-1 in the western AP (Van Wessem et al., 2016), while snowfall can be as low as 8 cm yr-1 in the interior of the East Antarctic ice sheet (EAIS) (Picard et al., 2019).

p3 L73 "α a newly introduced tuning parameter that will be changed as an experiment" : can you explain what does this tuning parameters mean physically? (Because I don't understand why you tune the initial radius and not e.g. the time rate τ)
We chose to introduce this tuning parameter on the initial radius instead of the time rate, because we wanted to target specifically the very fast metamorphism rate that is accompanied with very small grain sizes (see Fig. 1). If one would multiply the metamorphism rate with α, it would result in a lower metamorphism rate for coarse grains as well, which is not desired.
P5 L137
This current parameterization, however, is not optimized for Antarctic conditions, as the observations by Legagneux et al. (2004), on which the parameterization is based, were measured in the French Alps.

The temperature of the snow samples are relatively high compared to typical Antarctic temperatures, between 0 and -5.6 degrees C and they were stored in -15 degrees C. As snow metamorphism is faster for higher temperatures, the snow metamorphism scheme is therefore not directly applicable to the AIS. Hence, in the next experiment…

p4 L105-106 : "others contribute to mass loss, i.e., sublimation (SU), drifting snow erosion (ER) (...)" I don't understand why in the SMB definition drifting snow only contributes to mass loss, and why only mentioning drifting snow "erosion" and not drifting snow transport? Drifting snow transport can lead to mass convergence and thus snow deposition.
This is indeed confusing, changed the following to address this:
P4 L104
Some surface processes contribute to mass gain, i.e., snowfall (SN), rain (RA) and drifting snow accumulation, and others contribute…
P4 L106
…runoff (RU). In case of drifting snow accumulation, ER is negative. RU includes…

p5 L131 "a refreezing grain size of 1 mm". You did not defined this parameter and how it affect the snow metamorphism. → after a second reading I understand that it is the grain size given to the frozen water when it refreezes, seems now evident (...) but maybe you can just explicit it.
We changed the following:
P5 L131
…and a grain size of refrozen water of 1 mm. In GS, we kept the SLED at 5 mm as has been used for the Greenland ice sheet simulations of…

p5 L138-140 "SSA is increased from 60 to 100 m2 kg−1, reducing r from 55 µm to 37 µm. An SSA of 100 m2 kg−1 better matches observations of fresh snow at Dome C (Libois et al., 2015). Furthermore, this changes the dry snow metamorphism rate from the fastest to the slowest regime, reducing snow growth by an order of magnitude (Fig. 1)." : Why doesn't Fig. 1 show SSA greater than 81.8?
The lines in Figure 1 do start at 100 m2 kg-1 SSA, which is roughly equivalent to 37 micrometer. But we chose to show the grain radius 'r' on the x-axis, as it is for many people a more intuitive parameter than SSA, and we use round numbers for readability of the figure. These values are then converted to an SSA on the second x-axis, hence they are inevitably not rounded.

p5 L137-138 "This current parameterization, however, is not optimized for Antarctic conditions.". On which basis do you state this?
In preliminary studies we noticed that the SSA is way too low in Antarctica compared to Picard et al. (2016) and Libois et al. (2015) (See also Fig. 5), especially for very small grain sizes that are prevalent in most of the AIS. Furthermore, the measurements of Legagneux et al. (2004), on which the metamorphism parameterization is based on, were done under relatively high temperatures in the French alps (samples were measured between 0 and -5.6 degrees C and were stored in -15 degrees C). As snow metamorphism depends on temperature, these high temperatures compared to prevalent temperatures on Antarctica, results in a much stronger snow metamorphism than one would expect under the cold conditions of Antarctica. We changed the following
P5 L137
This current parameterization, however, is not optimized for Antarctic conditions, as the observations by Legagneux et al. (2004), on which the parameterization is based, were measured in the French Alps. The temperature of the snow samples are relatively high compared to typical Antarctic temperatures, between 0 and -5.6 degrees C and they were stored in -15 degrees C. As snow metamorphism is faster for higher temperatures, the snow metamorphism scheme is therefore not directly applicable to the AIS. Hence, in the next experiment…

p6 L155-156 "Modeled SMB is compared with 1870 SMB measurements including isolated observations and traverses on the EAIS (Fig. 2b). Favier et al. (2013) describe this data set in more detail." : Wang et al (2021 https://essd.copernicus.org/articles/13/3057/2021/essd-13-3057-2021.html) nicely updated this dataset.

Thank you for suggesting this new data set. We now use this data set and updated Fig. A1 and the locations in Fig. 2b. In addition, the following has been changed:

Fig A1

[Figure]

P6L155

Modeled SMB is compared with 1924 SMB measurements including isolated observations and traverses on the EAIS (Fig. 2b). Wang (2021) and Wang et al. (2021) describe this data set in more detail.

P17L311

Finally, compared to 1924 SMB observations in the EAIS (locations shown in Fig. 2b), the difference between CON and GS is small and both agree well with measurements (Fig. A1), with a bias of 23.5 and 24.4 mm w.e. yr-1, and RMSE of 106.4 and 106.0 mm w.e. yr-1, respectively. The correlation coefficient is 0.41 for both CON and GS.

p6 L158 "to specifically produce a melt rate estimate for Neumayer station (Fig. 2b)" : replace "(Fig. 2b)" by "(see location at Fig. 2b)" (I was expecting melt rates at Fig. 2b). Idem p8 L172 "(Fig. 2b)"

Done

Figure 2, Figure 3, Figure 10, Figure 11 : Add a label(title) on each panel for readibility, e.g. Fig. 2a) "GRL - Rp2", Fig2b) "CON - Rp2"

Added as requested

P9 L183 "In summer (not shown), the signal of Fig. 2 is amplified." : Show it in supplement.

We added a figure to the supplement with summer mean monthly-averaged T2m difference with Rp2 for GS and CON:

[Figure]

P9 L183-186 "A comparison with observations in DML during summer (Table 2), which is the season where any changes in the albedo have the strongest impact on the SEB, shows that the temperature of Rp2 is modeled well, with a small bias of -0.3C and a root-mean-square error (RMSE) of 1.4C. The bias of GRL and CON are larger: 2.0C and -0.8C respectively." : the table is nice but I am very frustrated by these numbers averaged over the 9 stations. I would like to see a map of the biases during summer (i.e. these same numbers but with colored dots, with one map for each of the main variables) to estimate the spatial variability in these biases. If not in the main text, it should be added in supplementary.

We have added a new figure where we show the bias of the T2m of the various stations on a map. We added this figure in the supplementary.

[Figure]

P9L186
…-0.8 degrees C respectively. For more inland stations like station 8, 9 and 12, the bias of GS is larger compared to stations close the edge of the ice sheet, while the bias of Rp2 and CON is smaller. This illustrates…

P11 L214-217 "Here, we show that the snow temperatures at Dome C (Fig. 4) match better with observations (Brucker et al., 2011) in CON than in Rp2 and GRL. During summer (Fig. 4a and b), we observe that Rp2 is somewhat too cold compared to measurements. Results improve for CON,

showing the significance of subsurface heating, although the skin temperature in DML is somewhat underestimated (Table 2)." : I think "Improvement" from Rp2 to CON shown in Fig. 4 should be interpreted with caution : (1) observed time series are very short (only one year); (2) observations are only at one location, Dome C; (3) on these profiles Rp2 and CON are very similar, meaning the "improvement" can be by chance. Furthermore, "Results improve for CON, showing the significance of subsurface heating" → there are many difference between CON and Rp2, why do you state that the main processes involved here is the subsurface heating ? E.g. in Fig4a) the "improvement" seems to be driven by warmer surface temperature with similar sub-surface gradients?

You are right that we jumped to conclusions too soon here. We have changed this text to the following to address the issues:

P11 L214

Here, we show that the snow temperatures of CON match well with observations  (Brucker et al., 2011) at Dome C (Fig. 4). During summer (Fig. 4a and b), we observe that Rp2 is somewhat too cold compared to measurements. The snow temperatures of GS are significantly overestimated by up to 10 C. During autumn (Fig. 4c), temperature profiles of Rp2 and CON, and to a lesser extent GS, are more similar, as surface temperature differences are smaller and the impact of radiation penetration diminishes towards winter (Van Dalum et al., 2021b). Compared to observations, however, temperatures in autumn are too high for this particular year.

Figure 5 : add all the experiments (or at least add them in supplementary)

Unfortunately, only Rp2, GS and CON have an overlapping time frame with the in-situ measurements. As we had limited computational time, we were unable to run all experiments for the full time period, hence we are unable to plot them here.

p12 L230-231 "The CON settings somewhat underestimate snow metamorphism, leading to higher SSA during summer, but this can be fine tuned using α in Eq. (1)." : Can you detail on this? Show how SSA is evolving for different α between 0.25 and 1? Or at least for your different experiments.

Figure 1 illustrates the way that the SSA evolves in the various experiments. It also shows the 'FSG' and 'FSM' experiment explicitly. This means that it already shows the case of $\alpha = 1$ (for FSG) and $\alpha = 0.25$ (for FSM). Changing $\alpha$ would therefore lead to a line in the figure between these two experiments. We added the following to clarify this:

P12 L231

Increasing $\alpha$ results in an SSA evolution, depending of the choice of $\alpha$, to be between FSG (which uses $\alpha = 1$) and FSM (which uses $\alpha = 0.25$) in Fig. 1.

Section 5 : as said above, I would like to see maps of these biases for the different fluxes, at least in supplement.

As requested, we have added a figure where we show the bias of the SEB components in a map on DML. Each circle chart represents an AWS, and each circle is split into three parts representing Rp2, GS and CON. We have added this figure in the main text. We also discuss it:

P12L239

Bias differences between most stations are limited, especially close to the edge of the ice sheet (Fig. [new figure number]a,b). For station 12 that is located on the Antarctic Plateau, both LWd and LWu are overestimated for Rp2 and CON.

P12L244

Similar to longwave radiation, the biases of most stations are similar, except for station 12, where SWd and SWu are underestimated (Fig. [new figure number]d,e).

P12L246

The SHF overestimation is stronger for station 16 and for more inland stations like station 8 and 12 (Fig. [new figure number]f). This can be either…

P12248

Similarly, GS also shows a better LHF representation than Rp2 and CON (Table 2 and Fig. [new figure number]c). Hence, turbulent…

[Figure]

Figure 7a) The divergent colormap is not relevant for showing the continuous albedo variable. Change the colormap for a continuous one.
See review #2.

Figure 8.g) "SSA as a function of depth" : modeled by CON ? + Rainbow colormap shoulb be prohibited, use a continuous colormap instead (e.g. viridis on python, parula on matlab).
Yes it is as modeled by CON and we added this in the caption. We have also changed the colormap.

[Figure]

p15 289-290: "Nonetheless, despite a slightly overestimated albedo, CON provides a better representation of the near-surface climate, albedo and near-surface snow state than the GRL experiment." : This sentence is not very convincing. Fig8h) suggests that albedo is better represented in GRL than in CON. So at the end, in Rp2 and CON you have a better air temperature but for the wrong reasons... However, your conclusions are for Neumayer only, you should highlight that in this sentence. The best would be to compare maps of the different SEB components vs. observations, to have a broader view on the models biases for different stations (as asked above for Section 5).

Indeed, not all parts of this paragraph are formulated correctly. Hence, we have rewritten the following:

P15 L288

The analysis of the SEB shows that there are some compensating biases, i.e., clouds and turbulence. Despite on average only minor albedo changes between CON and Rp2 (Fig. 6), we also show by analyzing a case study for Neumayer that with the introduction of a new physically based snow albedo and radiative transfer scheme the instantaneous albedo can differ considerably. In particular radiation penetration and spectral shifts due to cloud cover and high SZA lead to high day-to-day albedo variability.

P18 L322-324 : "For the time step currently employed in Antarctic simulations, a SLED of 5 mm leads to slightly overestimated heat buffering in the uppermost part of the snow layer, leading to more internal melt." : On which basis do you ground this statement? From what I can understand, you tuned all your parameter to obtain the closest results to Rp2? But how are you sure that Rp2 it the closest to reality? (after reading the next section, the comparison with QuickSCAT, it's more convincing).

By applying the scale analysis that can be found in the appendix of Van Dalum et al. (2021), one can determine that the estimation of the SLED of 5 mm is on the lower end for this study. And a lower SLED results in more heat buffering in the upper layers (See Fig. 1 of Van Dalum et al. (2021)) and more energy available for internal melt. We changed the following in the text:

P6 L144

…model time step of 6 minutes. This adjusted SLED takes away the slight overestimation of subsurface heating introduced by using a SLED of 5 mm.

P18 L322

For the time step currently employed in Antarctic simulations, a SLED of 5 mm is on the lower end of the scale analysis that is employed in Van Dalum et al. (2021). This underestimation of the SLED results in a slightly overestimated heat buffering in the uppermost part of the snow layer, leading to more internal heat absorption. Hence, melt is expected to be further reduced by increasing the SLED. This is indeed the case when comparing RFG (Fig 10d) with CON (Fig 10e), illustrating the impact of subsurface heating.

Figure 11a) you use a divergent colormap for a continuous variable (melt), you should use a continuous colormap instead (e.g. viridis on python, parula on matlab).
See previous review

p19 L340-341 : "This shows the impact of internal heating and melt-albedo feedback, as these processes significantly enhance melt, similar to findings of Jakobs et al. (2019)." : It can just be an albedo effect (in absolute value), not necessarily a "internal heating and melt-albedo feedback" effect. Why do you advocate for the later?
We reformulated this part to address the issue:
P19 L339

Integrating melt over the AIS shows a similar pattern (Fig 12), with melt in GS almost an order of magnitude larger than QSCAT in every year. Rp2, on the other hand, compares well with observations. The addition of a new snow albedo and radiative transfer scheme in Rp3 impacts the strong melt-albedo feedback, similar to findings of Jakobs et al. (2019), enhancing melt. Differences with QSCAT are reduced when all changes of the sensitivity experiments are implemented, leading to a better correlation with CON. The interannual variability compares well for all experiments.

p20 L349-352 : "In conclusion, melt of the AIS is somewhat sensitive to fresh snow SSA and fresh dry snow metamorphism and is highly sensitive to the refreezing grain size and SLED. Hence, subsurface heating can warm the snowpack considerably, enhancing melt. Despite the low average melt flux in Antarctica, the impact of subsurface heating should not be neglected for a physical description of (sub)surface melt." : This conclusion does not fit in this section as you discuss only CON and GRL here. In fact this is the same remark as just above. If you want to highlight the impact of "subsurface heating", you should specifically show the simulations where you only change parameters affecting sub-surface heating, and not the fresh snow albedo; e.g. SLED if I understand well, and maybe the refreezing grain size, but this latter may one also affect the surface albedo (?)
We agree that this part does not fit in this paragraph. We have moved the following and removed the rest of this paragraph:
P21 L378

…refreezing grain size and SLED. The difference between RFG and CON illustrates the importance of subsurface heating, which can warm the snowpack and enhance melt. Despite the low average melt flux in Antarctica, the impact of subsurface heating should not be neglected for a physical description of (sub)surface melt. It is clear that…

p21 L371-373 : "These biases demonstrate that Rp2 and CON provide a better representation of the surface climate than GRL" : I don't understand why?
We have rewritten this part and added a part about the albedo:
P21 L370

Analysis of the SEB shows that Rp3 exhibits, on average, some small (lower than 10 W m-2) persistent biases in the net radiative fluxes, which is caused by too transparent clouds and overestimated

turbulent surface fluxes. This illustrates that there is still room for model development, especially in the turbulent fluxes. With the introduction of a new physically based albedo and radiative transfer scheme, more processes now impact the snow albedo. Radiation penetration and spectral shifts due to cloud cover and high SZA can lead to albedo differences up to -0.1 between CON and Rp2. Monthly-averaged, however, differences between these model versions are small.

p21 L374-375 : "The higher (subsurface) temperatures in GRL lead to excessive melt around the margins and on the ice shelves, locally leading to runoff and a reduced SMB." : Why "subsurface"? Can't it be only caused by surface temperature increase?

As GS also includes subsurface heating, it is unlikely that it is only caused by a higher surface temperature. Moreover, we show that with increasing the SLED, which impacts energy available to heat the subsurface temperature, less melt is modeled (so the difference between RFG and CON). So higher subsurface temperatures will impact the surface temperature and hence melt.

P21 L381-382 : "In conclusion, introducing a more physically based albedo scheme in RACMO2 that allows for radiation penetration and subsurface heating improves, after tuning, subsurface snow temperatures in Antarctica." In Section 2.1 you explain that the difference with Rp2 is the full albedo scheme, that has been changed for TARTES adapted for RACMO2 (with SNOWBAL). Here you only cite changes in "radiation penetration and subsurface heating", why citing these two changes only?

We do also mention that we introduce a more 'physically based albedo scheme', and explicitly say that it also includes radiation penetration and subsurface heating. So we also meant the full albedo scheme. To remove the confusion, we reformulate it to the following:
P21 L381
In conclusion, by introducing a new more physically based spectral snow albedo and radiative transfer scheme in the polar version of RACMO, which also allows for subsurface heating, improves, after tuning, the subsurface snow temperatures in Antarctica. Incorrectly modeling snow...

---

## Author Response (AR2)

Referee and editor comment responses on the manuscript:
*Sensitivity of Antarctic surface climate to a new spectral snow albedo and radiative transfer scheme in RACMO2.3p3*
by C.T. van Dalum et al.

Again, we would like to thank the referees and editor for their comments. Similar as before, the comments are in black and the response in orange.

**Report #1**
I totally disagree on this and I think there is a misunderstanding. I think the main differences between the maps shown in the responses is that when they used continuous sequential colormaps (left) and their original diverging colormaps (right), they also used continuous increments on the left whereas they used discrete increments on the right. Discrete increments are very good and recommended for data visualization. Without any doubt authors must use a continuous colormap such as viridis for continuous variables, everywhere in the article, but with discrete increments similar as in their original figures.
Indeed, there has been a misunderstanding. We thought that you wanted us to use both a continuous colormap as well as continuous increments. As requested, we have updated the colormap to a continuous one with discrete increments for all relevant figures, i.e., Fig. 8a, Fig. 9 and Fig. 12a.

**Editor Comment**
According to reviewers, your paper is now ready to be accepted as you dealt with all their remarks except the remark of Referee #3 about the colour scheme used for some figure. I agree with her that a blue - yellow - red palette cannot be used here (as it is reserved for a centred palette on zero). Could you suggest a new "continuous" palette for these figures as requested by Referee #3?
As requested, we have updated Fig. 8a, Fig. 9 and Fig. 12a with a continuous colormap but with discrete increments.

Finally, I suggest to add the comment in brackets in line 412:
... allows for subsurface heating, improves, after tuning (as biases were partly compensated in former RACMO versions), the subsurface snow temperatures in Antarctica...
We have added this comment in brackets, as requested.

Furthermore, a couple of minor textual changes have been implemented on p1 L6 and p21 L379.